# Motility of an autonomous protein-based artificial motor that operates via a burnt-bridge principle

Chapin S. Korosec [1,6,7] ✉, Ivan N. Unksov[2,7], Pradheebha Surendiran[2], Roman Lyttleton[2], Paul M. G. Curmi [3], Christopher N. Angstmann [4], Ralf Eichhorn [5], Heiner Linke [2] ✉ & Nancy R. Forde [1] ✉

Inspired by biology, great progress has been made in creating artificial molecular motors. However, the dream of harnessing proteins – the building blocks selected by nature – to design autonomous motors has so far remained elusive. Here we report the synthesis and characterization of the Lawnmower, an autonomous, protein-based artificial molecular motor comprised of a spherical hub decorated with proteases. Its "burnt-bridge" motion is directed by cleavage of a peptide lawn, promoting motion towards unvisited substrate. We find that Lawnmowers exhibit directional motion with average speeds of up to 80 nm/s, comparable to biological motors. By selectively patterning the peptide lawn on microfabricated tracks, we furthermore show that the Lawnmower is capable of track-guided motion. Our work opens an avenue towards nanotechnology applications of artificial protein motors.

Molecular motors are essential for powering directional motion at the cellular level, including transport and sorting of cargo, cell locomotion and division, and remodeling of the extracellular matrix[1,2]. Biological molecular motors are made of proteins whose directed motion is autonomously coupled to the consumption of chemical free energy. Inspired by such biological machines, significant strides have been made, using small synthetic molecules or DNA as building blocks, to design and implement synthetic devices capable of directed motion on the nanoscale[3–13] and the microscale[13–16]. With a notable exception[8], these designs have not achieved motion along lengthy one-dimensional tracks, akin to cytoskeletal and extracellular filaments and a prerequisite for many potential nanotechnological applications.

Novel motor proteins have been crafted by swapping or re-engineering domains of protein motors[17–23]; however, these have been based on naturally occurring motors. Distinct ideas have been proposed for using nonmotor proteins as building blocks to engineer motors[13,21,24–27], but to our knowledge, motility of a synthetic motor constructed of proteins – the material system that enables the complexity of life – is an important ongoing challenge. Creating a synthetic motor and its track entirely of protein components, which themselves lack motor properties, would demonstrate the abilities of proteins to be used as components of a bioengineering toolbox, and would represent a significant step forward in the field of synthetic biology.

Here, we demonstrate biased motion of a protein-based synthetic motor dubbed the Lawnmower. Of the many possible approaches for achieving directed motion at the nanoscale, the Lawnmower is designed to operate as a burnt-bridge Brownian ratchet (BBR)[28], whereby it exploits chemical free energy and polyvalency to achieve directed motion on an underlying substrate. Removal of surface-bound substrate sites by a BBR induces a local free energy gradient, which biases the BBR's thermally driven motion towards energy-rich intact substrate sites. Many different biological systems undergo directed motion as BBRs. These include the *Influenza* virus[29–31], bacterial plasmid partitioning[32] and bacterial engulfment[33], and the ligand-

[1]Department of Physics, Simon Fraser University, Burnaby, BC V5A 1S6, Canada. [2]NanoLund and Solid State Physics, Lund University, Box 118, SE – 22100 Lund, Sweden. [3]School of Physics, University of New South Wales, Sydney, NSW 2052, Australia. [4]School of Mathematics and Statistics, University of New South Wales, Sydney, NSW 2052, Australia. [5]Nordita, Royal Institute of Technology and Stockholm University, 106 91 Stockholm, Sweden. [6]Present address: Department of Mathematics and Statistics, York University, Toronto, ON M3J 1P3, Canada. [7]These authors contributed equally: Chapin S. Korosec, Ivan N. Unksov. ✉e-mail: chapinskorosec@gmail.com; heiner.linke@ftf.lth.se; nforde@sfu.ca

depleting migration mechanism of metazoan cells[34]. The BBR mechanism appears to be the mechanism of choice for regulated enzymatic degradation of key extracellular scaffolds in animals and plants, including collagen-degrading mammalian matrix-metalloproteases (MMPs)[35,36], cellulase[37,38] and chitinase[39,40]. Synthetic BBRs have been created that use DNA to achieve unguided, two-dimensional motion at the nanoscale[6–10,41] and microscale[14–16]. However, to our knowledge, long-range track-guided motor activity of a polyvalent BBR has not yet been achieved.

In this work, we characterize the dynamics of the Lawnmower. We find that the Lawnmower exhibits biased motion on a two-dimensional

peptide lawn, and further implement the Lawnmower on one-dimensional peptide lawns microfabricated to produce track-guided motion of this protein-based synthetic design.

## Results

As shown schematically in Fig. 1A, the Lawnmower (LM) consists of multiple trypsin proteases coupled through a central microspherical hub; these protease "blades" bind to and cleave peptide substrates presented via an F127 polymer brush adhered to a surface[25,42] (Methods: Lawnmower synthesis and Lawn preparation). Once the LM lands on the surface, diffusion of the central hub allows protease blades to

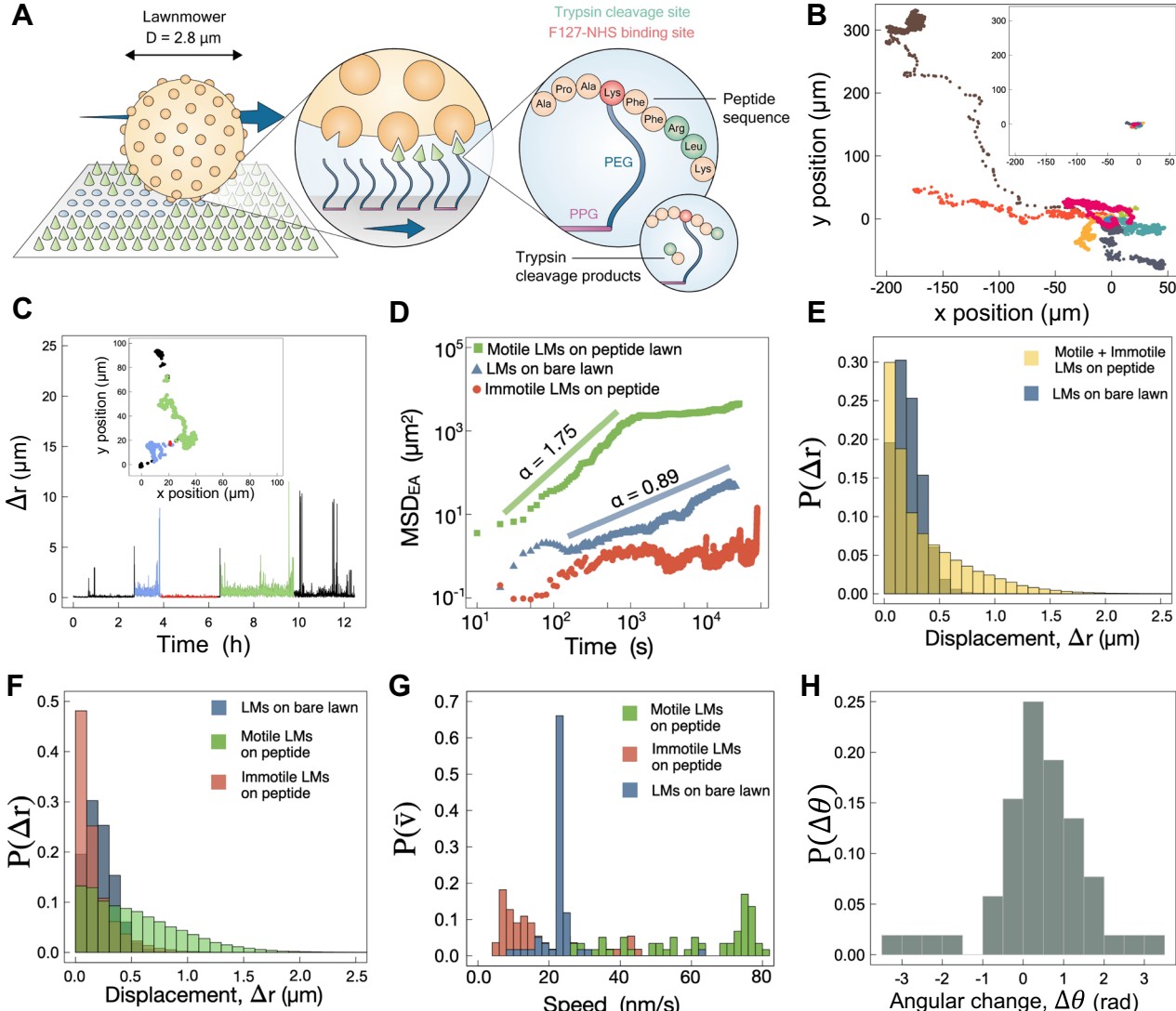

**Fig. 1 | Lawnmowers exhibit motor-like dynamics on peptide lawns. A** 2D schematic of the LM mechanism. The Lawnmower is designed to move as a burnt-bridges ratchet (BBR) via the successive cleavage of surface-bound peptides (green cones), leaving a wake of product (blue circles). Peptide is linked to the surface via lysine to NHS-end-modified F127 (PEG-PPG-PEG). **B** Example LM trajectories on a peptide lawn spanning 6.25 hours of motion. (Trajectories of the full 12.5 hour range are shown in Supplementary Fig. 2.) Trajectories are depicted as starting from a common origin. Inset: example LM trajectories on a bare lawn, also spanning 6.25 hours, travel significantly shorter distances. For an expanded view of their diffusive motion, see Supplementary Fig. 3. **C** Step size $\Delta r$ vs time throughout one 12.5-hour LM trajectory (green in **B** and Supplementary Fig. 2) on a peptide lawn. $\Delta r$ is determined for each 10-second time interval. Dynamics are saltatory, with bursts of activity interspersed with lengthy immotile dwells. The average speed of this LM is $\bar{v} = 36$ nm/s. Different regions have been colored based on visual inspection, as a

guide to the dynamical heterogeneity within a trajectory. **D** $\text{MSD}_{\text{EA}}(r)$ as a function of time for motile (green squares, $n = 59$), and immotile (red circles, $n = 55$) LMs on peptide lawn, as well as LMs on bare lawn (blue triangles, $n = 59$). **E** Distributions of displacements over all 10-second intervals for all LMs on peptide and bare lawns. **F** Displacements from (**E**) are plotted separately for motile and immotile classes of LMs on peptide lawns. Small displacements of motile LMs correspond to periods of immotility within the trajectory (e.g. **B**). **G** Distributions of trajectory-averaged mean speeds of all individual motile and immotile LMs on peptide and bare lawns. LMs on bare lawns are generally much slower, with a distribution of mean speeds centered around an ensemble-average mean speed of $\langle \bar{v} \rangle = 23 \pm 4$ nm/s. $\langle \bar{v} \rangle$ of motile LMs on peptide lawns is significantly higher: $\langle \bar{v} \rangle = 58 \pm 20$ nm/s. **H** Consistent with the expected BBR mechanism, long steps (here, $\Delta r > 10$ μm) are directionally persistent, as shown by the peak in the distribution of angular changes around $\Delta \theta = (\Delta \theta_{i+1} - \Delta \theta_i) = 0$.

engage nearby peptides in any direction. The initial cleavage and diffusion direction breaks symmetry. The lack of fresh peptides behind the LM creates a free energy gradient and biases diffusion of the LM towards uncleaved peptide "grass" (Fig. 1A, Supplementary Fig. 1). We expect to see LM dynamics on a two-dimensional peptide lawn exhibiting this biased motility (similar to a self-avoiding walk), whereas on a lawn lacking peptides (referred to as a bare lawn), the LM is expected to exhibit diffusive dynamics[43].

We verified the activity of LMs in solution and compared their peptide cleavage rate to that of trypsin (Supplementary Fig. 1). We estimate that LMs each accommodate $(5 \pm 1) \cdot 10^5$ active trypsins, or about 0.02 trypsins per $nm^2$ of the surface area of the microsphere. Of these, ~2000 trypsins can be engaged with the underlying peptide lawn, which presents ~$10^4$ peptides within this LM footprint (Supplementary Note 1). Thus, LMs are active and highly polyvalent, with the potential for thousands of interactions with the lawn.

We observed striking differences between trajectories of LMs on a peptide lawn (peptide-F127) and on a peptide-free, bare lawn (F127) (Fig. 1B, Supplementary Figs. 2 and 3). Qualitatively, it is immediately apparent that over similar timescales LMs on peptide lawns travel much farther than on bare lawns. Furthermore, directional bursts can be seen on the peptide lawn, which are not observed on the bare lawn. Thus, the observed LM dynamics on peptide lawns can be described as saltatory[15,44,45], in which bursts of long-range travel are interspersed with highly localized, quasi-immotile dynamics (Fig. 1C, Supplementary Fig. 2 and Movie 1). This contrasts with the time-invariant dynamics of LMs on bare lawns, in which essentially all LMs ( < 1% immotile) exhibited diffusive motion (Supplementary Fig. 3). In fact, while the majority of LMs exhibited motile dynamics on peptide lawns (52%; number of LMs $n = 59$), a significant fraction remained immotile throughout the 12.5-hour experiment ($n = 55$). We classified a LM as "motile" if its mean-squared displacement exceeded a threshold value of 10 $\mu m^2$ at $\tau = 4400$ s, and "immotile" otherwise (Supplementary Fig. 4A). To correct for sample drift, trajectories of immotile LMs were averaged and this average trajectory was subtracted from each motile LM trajectory (Supplementary Fig. 4B). Our characterization of LM dynamics focuses on these drift-corrected trajectories of the motile class of LMs: while they may contain periods of immotility ranging from minutes to hours, they all contain active periods of motility, and the entire trajectories were analyzed without internal segregation. Later, we discuss possible reasons for immotility, observed only on peptide lawns.

To quantitatively assess the dynamics of the Lawnmowers on the two types of surfaces, we first followed an approach used for other microscale BBRs[15,16] and determined how the mean-squared displacement (MSD) of LMs evolves with time, where MSD $\propto t^\alpha$. The scaling of MSD with time characterizes the type of diffusion[46]: subdiffusion, conventional diffusion, and superdiffusion correspond to $0 \leq \alpha < 1$, $\alpha = 1$, and $1 < \alpha < 2$, respectively, while for a purely ballistic system proceeding at a constant velocity, $\alpha = 2$. From this picture, LMs with BBR dynamics on peptide lawns should exhibit $\alpha > 1$, while LMs on bare lawns should display conventional diffusion with $\alpha \approx 1$. We use ensemble-averaging and trajectory-averaging to analyze the MSD, with dynamics characterized by $\alpha_{EA}$ and $\alpha_{TA}$, respectively (Eqs. 4, 5). We find that LMs exhibit dynamics consistent with their designed function: trajectories of motile LMs on a peptide lawn are characterized by strongly superdiffusive, ensemble-averaged dynamics at early experimental measurement times ($\alpha_{EA} = 1.8$ over tens of minutes), in contrast to the diffusive-like dynamics of LMs on bare lawns ($\alpha_{EA} \lesssim 1$; Fig. 1D). The isotropic distribution of trajectories at early times confirms that this large value of $\alpha_{EA}$ is not due to sample drift (Supplementary Fig. 5).

A distinct, trajectory-averaged approach to MSD analysis (MSD$_{TA}$, calculated as a function of time lag $\tau$ using Eq. 5) further reveals the distinct dynamics of LMs on the two surfaces: on peptide, LMs are characterized by a range of MSD$_{TA}$ values, contrasting strongly with the uniformly diffusive dynamics of LMs on bare lawns (Supplementary Figs. 4, 6). Comparison of the average anomalous diffusion exponent from the ensemble of MSD$_{TA}$ indicates that motile LMs retain, on average, superdiffusive dynamics across their entire measured trajectories of 12.5 hours with $<\alpha_{TA}> = 1.1$ for motile LMs on peptide lawns, while LMs on bare lawns are characterized by $<\alpha_{TA}> = 0.95$ (Supplementary Fig. 7), consistent with diffusive behavior. We expect that $<\alpha_{TA}> = 1.1$ underestimates the superdiffusivity of active LMs because this treatment assumes time-invariant dynamics and averages out the bursts of active motion with the dwells of immotility observed in trajectories on peptide lawns[45]. Importantly, the difference between MSD$_{EA}$ and MSD$_{TA}$ indicates that the system is nonergodic[46]. This is consistent with the history-dependent dynamics of LMs: their trajectories are influenced by what regions of the peptide lawn they have previously visited and cleaved.

The enhanced motility of LMs on peptide lawns is also revealed by the probability distribution of their displacements $P(\Delta r)$ and corresponding speeds $v$, measured within each $\Delta t = 10$ s recorded time interval. For LMs on bare lawns, displacement distributions are well described by the Rayleigh distribution for diffusion in 2D and provide expected diffusion coefficients for microscale LMs (Supplementary Fig. 8 and Supplementary Note 2). (Gravity holds the LMs against the lawn; the observation of 2D diffusive motion is consistent with the previously demonstrated ability of the F127 surface to block non-specific adhesion of these microspheres[42].) By contrast, LMs on peptide lawns have heavy-tailed displacement distributions that do not agree with a Rayleigh distribution (Fig. 1E, F). The heavy-tailed displacement distribution of LMs on peptide lawns translates into an approximately three-fold higher mean interval speed of the motile ensemble ($<v> = 58 \pm 20$ nm/s), compared to LMs on bare lawns ($<v> = 23 \pm 4$ nm/s) (errors are standard deviations).

Alternatively, for each LM we determined its average interval speed throughout its trajectory, $\bar{v} = \overline{\Delta r}/\Delta t$, where the average displacement per $\Delta t = 10$ sec interval $\overline{\Delta r}$ is given by Eq. 3. Even though LMs exhibit extended immotile dwells during their trajectories (Fig. 1C) that decrease their average speed $\bar{v}$, nonetheless there is a clear distinction between LM speeds when fueled by peptides versus on a bare lawn (Fig. 1G). LMs on bare lawns exhibit homogeneous dynamics, possessing trajectory-averaged speeds tightly clustered around $\bar{v} = 23$ nm/s. By contrast, trajectory-averaged speeds of individual motile LMs on peptide lawns are more broadly distributed and are generally much faster, with 50% of motile LMs having trajectory-averaged speeds of $\bar{v} = 75 \pm 5$ nm/s. The broad distribution in mean speeds is consistent with the heterogeneity in dynamics on peptide lawns revealed by other displacement-based metrics (Fig. 1C, Supplementary Figs. 2, 4). Although diffusion may be an effective means of exploring local space at short timescales, the coupling of chemical energy to directed motion allows molecular motors to travel much farther at longer timescales than is possible under thermal diffusion. We see this motor-associated greater range of motion for peptide-fueled versus diffusing LMs clearly demonstrated even at our shortest observation times of 10 seconds, where the average distance travelled is $<\overline{\Delta r}> = 580 \pm 200$ nm on peptide lawn versus the diffusive $<\overline{\Delta r}> = 230 \pm 40$ nm on bare lawns.

Performance measures of our protein-based LMs can be compared with DNA-based microscopic BBRs. The microsphere-based Par design of Vecchiarelli et al. represents an in vitro reconstitution of the natural BBR-type plasmid partitioning system in E. coli[15], and had an average speed of 100 nm/s. Somewhat lower average speeds of 30-50 nm/s were achieved by highly polyvalent DNA motors (HPDMs), a synthetic microscale BBR system[16]. These speeds are comparable to the average speed of $\approx 60$ nm/s exhibited by motile LMs on peptide lawns. The appearance of trajectories differs among these systems: Par trajectories were generally quite straight, while HPDMs moved in a less directed fashion. In our LMs, active bursts on peptide lawns appear

directional (Fig. 1B; Supplementary Movie S), while the more localized dynamics appear more isotropic. Quantifying this distinction, we found that active runs of LMs on peptide lawns are directionally persistent: a large step is likely to continue in the same direction as the previous step (Fig. 1H). This is consistent with a BBR mechanism. By contrast, short-range dynamics are directionally anticorrelated (Supplementary Fig. 9). Anticorrelated step directions can be caused by an effective restoring force presented by the polyvalent binding interactions between trypsins and peptides on the lawn. Simulations of spherical BBRs on 2D landscapes have shown that directional persistence is influenced by both the polyvalency and entropic stiffness of these motor-track binding interactions[43], which may account for some of the differences observed among these different realizations of microscale BBRs.

Track-guided motility is a ubiquitous mechanism of biological molecular motors[1]. Furthermore, guidance of cytoskeletal filaments along microfabricated tracks by biological molecular motors is being intensely explored for applications ranging from biosensing and diagnostics to energy-efficient computation[47–50]. For many such applications, it would be of great interest to develop custom-made protein motors with bespoke properties and capability of moving along a pre-defined track. Thus, following the demonstration of biased motion of LMs on a 2D surface, we investigated whether LM motion could be guided along narrow tracks. We used chemical specificity to deposit peptide lawns or bare lawns onto the bottoms of microfabricated channels (Fig. 2A)[51]. Gravity constrained the dense beads within these 0.5 μm deep channels, which provided effective 1D confinement. The channel width of 2.2 μm was chosen narrower than the LM diameter but sufficiently wide to retain polyvalency of LM-track interactions. Simultaneously tracking particles in orthogonal sets of channels (vertical and horizontal in Fig. 2B) allowed us to check for background flow.

We observed the dynamics of LMs to be guided by these quasi-1D tracks, both for LMs on peptide lawn and particles on bare lawn (Fig. 2B and Supplementary Movies 2–5). There is compelling evidence that LMs exhibit enzymatically driven motion in peptide channels, interspersed with diffusive motion and immotile periods (examples are shown in Supplementary Movies 3, 4). First, similar to the 2D case, LMs on tracks transition between motile and immotile states when on a peptide lawn (Fig. 2C) while particles on bare lawns exhibit time-invariant dynamics (Fig. 2D). Second, these LM displacement distributions $P(\Delta x)$ exhibit heavy tails, even when immotile dwells are excluded. This is seen by the heightened probability of large displacements for LMs on peptide lawns relative to a Gaussian distribution, whereas displacements in bare lawn channels exhibit the expected Gaussian distribution associated with 1D diffusion (Fig. 2E, F). The shape of the displacement distribution can be quantified by its kurtosis (Eq. 8), which indicates the tailedness of a distribution: for a Gaussian distribution, $\kappa = 3.0$. For LMs on peptide lawn channels, we find a kurtosis $\kappa = 3.3$, indicating larger displacements than expected from a normal distribution. Because this analysis truncated trajectories when they became immotile, the heavy tails of the distribution indicate motor-driven motility (Methods: Analytical methods and Supplementary Note 3). In contrast, particles on bare lawn have $\kappa = 3.0$, the expected value for one-dimensional diffusive motion.

To validate these observations of heavy-tailed distributions associated with motor activity, we developed a simple model according to which enzymatic LM action initially creates an area of bare lawn inside a channel. Subsequent LM action can take place only at the left or right edge of the bare area and is interrupted by free diffusion (for more details of this model see Methods: 1D Lawnmower model and Supplementary Fig. 10). In this model, the role of the observed small background flow is to bias LM action towards one side, consistent with the slight overall directionality apparent in Fig. 2B. Data modeled in this way show a kurtosis $\kappa = 3.2$, consistent with the heavy tails observed in experiments (Fig. 2E).

In contrast to this simple model, we do not expect that in the experiment a completely bare lawn is created at first contact with a LM. To check for this, we statistically analyzed the magnitude of LM displacements in the experiment conditioned on whether they occur near the edge of previously unvisited regions versus in previously visited regions (Supplementary Fig. 11). We find larger displacement near the edges, consistent with LM action (the same is not observed in control experiments with bare lawn), but we also observe large displacements in previously visited regions, indicating the presence of remaining lawn.

## Discussion
It is important to note that the periods of subdiffusive, quasi-immotile behavior that we observe both in 2D and in channels are especially likely in highly polyvalent systems such as the LM. Indeed, in some replicates LMs display much greater periods of immotility in peptide channels (Supplementary Fig. 12). Immotile dwells have been also observed for highly polyvalent DNA motors (HPDMs), where immotility was attributed to entrapment within previously visited, product-rich regions[16]. We, too, see evidence of immotility coinciding with previously visited regions: for example, the long dwell from 4-6.5 hours illustrated by the red color in Fig. 1C overlays with the earlier portion of the trajectory shown in blue (Fig. 1C, inset). Even small-length fluctuations would be sufficient to drive the LM into a previously visited region, which has the potential to lead to stalling due to the high polyvalency of (even very weak) trypsin-product interactions[52,53]. A second mechanism for temporary, often long-duration immotility in highly polyvalent systems is the long timescale associated with cleaving the large number of substrates under the footprint of the motor. In nanoscale HPDMs, release from such a state was posited to give rise to large displacements, associated with a heavy tail in the displacement distributions[10]. This mechanism was demonstrated in simulations of the reconstituted microscale Par BBRs, which found step size to inversely correlate with the polyvalency of bead-track interactions during trajectories[44]. It is likely the same dynamics are at play in LMs on peptide tracks, producing the observed saltatory dynamics and heavy-tailed displacement distributions (Figs. 1E, F and 2E, F).

Thus, although an increase in LM valency may lead to an increase in speed[10,54], when LM-track interactions are sufficiently high in number, the LM has a higher probability of becoming immotile, as its motion is quenched by the polyvalency of binding interactions[53]. Once enough bonds have broken, the LM is able to capture thermal fluctuations and rapidly access nearby peptides, and if cleavage is sufficiently rapid compared to the formation of many new binding interactions, then large-scale motion continues until enough interactions stochastically form to again trap the LM locally in an immotile state[44]. Such dynamics appear to be shared among emerging examples of BBRs, and warrant further investigations to optimize performance. Indeed, nucleic-acid-based BBRs of low polyvalency followed patterned tracks (over short distances[7] and over micrometers[8]), while optimization of motor-like motility along tracks for highly polyvalent systems such as HPDMs[16] and the LM remains a future challenge.

The precise mechanism responsible for motile and immotile states, observed generally for these BBR motors, remains an intriguing and important problem for the field to address. Starting from the initial performance described here, the LM design lends itself to modular design alterations that could be targeted in future studies to enhance and rationally explore this mechanism of motility. These include multivalency[25,53–58]; kinetic rates, particularly the appropriate balance of binding, cleavage and diffusion[32,43,44,58–61]; and linker length and flexibility[43,44,58]. Because the LM and track combine to enable the BBR mechanism, changes in the peptide lawn, such as the stiffness of the peptide supports and the surface density and dimensionality of the peptide track, can also be used to improve performance[43,59,62]. The

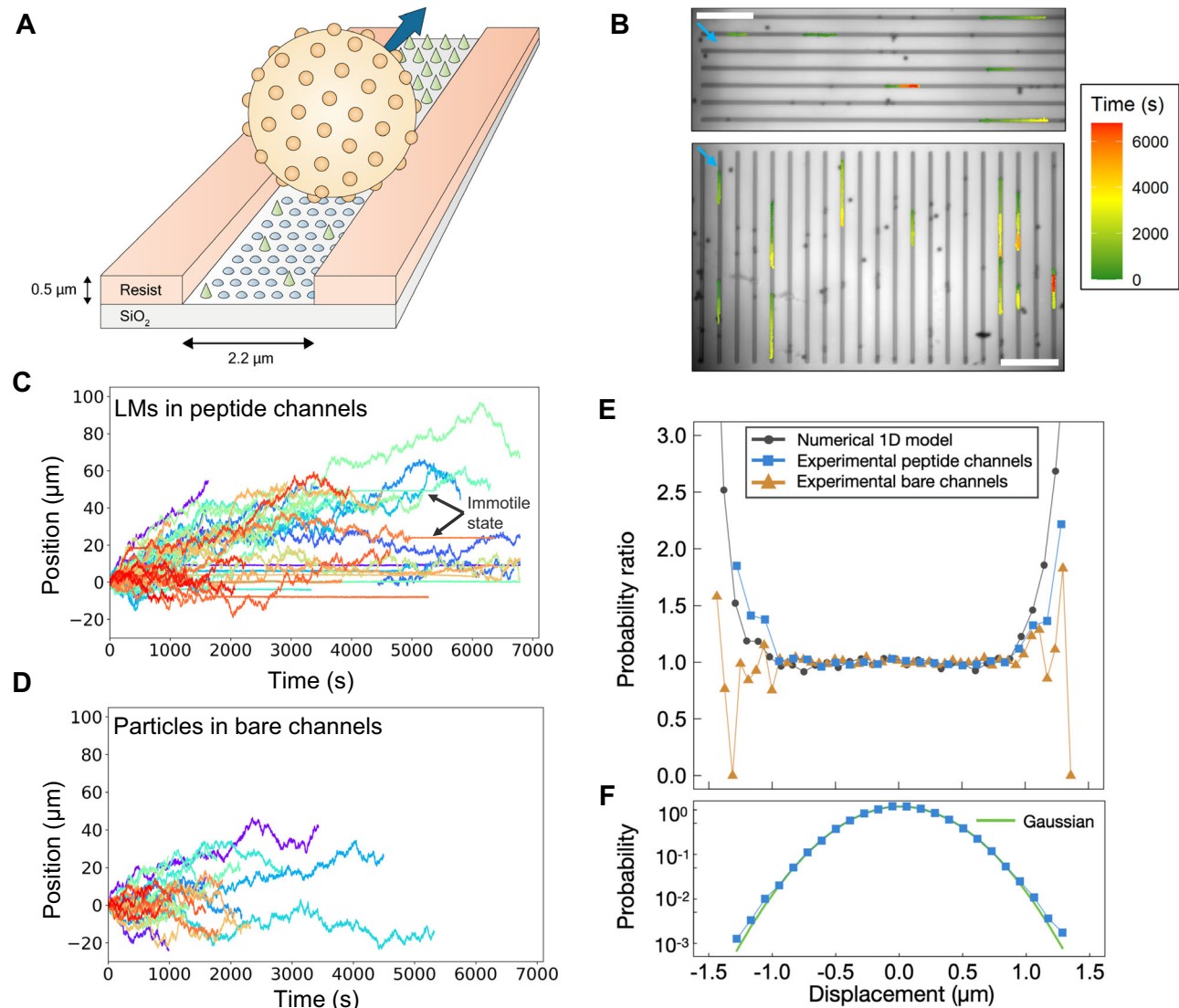

**Fig. 2 | Lawnmowers exhibit track-guided motion. A** Schematic illustrating LM motion within a lithographically defined peptide-containing channel with polymer resist walls and a SiO₂ channel floor that was selectively functionalized. LM beads have a density of 1.6 g/cm³ and are partially confined to the inside of channels by gravity. Channels are patterned in two directions as a control for possible background flow. **B** Trajectories of LMs on peptide lawns, colored from green to red to illustrate the time course of motion. Trajectories are overlaid with an image of the orthogonally patterned channels in the beginning of imaging (trajectories that started later are not depicted). The light blue arrows show the direction of slight background flow (≈ 0.01 μm/s). Scale bar is 50 μm. **C**, **D** Position along the channel direction as a function of time for (**C**) n = 48 LMs in peptide channels and (**D**) n = 26 particles in bare channels. The starting position of each LM is plotted with a

common origin, with the positive direction chosen to align with the observed background flow in the channel. **E**, **F** Displacement distributions relative to Gaussian distributions. **E** Comparison of experimental and model results to Gaussian distributions. Plotted are the ratios of probability densities for the measured versus Gaussian-predicted distributions, where the first and second moments of the Gaussian distribution match experiment. Blue squares: peptide lawn from (**C**). Orange triangles: bare lawn from (**D**). Black circles: 1D LM model. The model captures the heavy tails seen for LMs on peptide lawns, while bare lawn trajectories are described by diffusion. **F** Displacement probability distribution of LMs in peptide channels has heavier tails than a Gaussian distribution, as seen also by the kurtosis of the distribution of κ = 3.3.

microscale LM and its concomitant high polyvalency enabled straightforward experimental measurements, but inadvertently may have led to periods of highly localized dynamics. Reducing multivalency to the level where a protein hub[63] could be used to link a small number of trypsins could create an entirely protein-based BBR closer to the size scale used by nature. The demonstration that the LM can be guided by tracks opens the door to future experiments using bespoke properties to sort such motors, for instance, by patterning tracks in more complex geometries and with distinct peptide cleavage signals to guide LMs based on their constituent proteases.

Distinct from the related reconstituted Par and synthetic HPDM microscale BBR systems[15,16], the LM does not rely on the supply of

reagents from solution (Par proteins + ATP and RNaseH, respectively) to sustain motion. This offers the advantage of being a modular system that could be implemented in a variety of settings, without the need to maintain a supply of reagents. A drawback is that sustained LM dynamics require activity of its constituent trypsins; inactivated proteins cannot be replaced as easily as the solution-based supply in these other systems. However, in contrast to these solution-reliant BBRs, the LM system does not require external reagents to be provided to maintain activity. In this aspect, the LM bears more similarity to extracellular biological systems such as collagenases[35], cellulases[37,38] and chitinases[39], in which the free energy to power directed motion comes from cleavage of the track

itself. Like these biological BBRs, the LM operates autonomously to cleave and self-propel along its track.

While BBRs are inherently destructive to their tracks, one can envision pairing such motors to have guiders and followers, for example kinase- and phosphatase-based LMs that work cooperatively to guide each others' paths in complex environments. LMs could also become environmentally responsive, by incorporating pH- and temperature-dependent proteases in their design. Through evolution, nature arrived at proteins as its responsive, controllable building material; by harnessing proteins for synthetic purposes, we forge their potential application to a range of problems.

# Methods

## Lawnmower synthesis

The synthesis of microscale Lawnmowers was adapted from the previous protocol for quantum-dot-based LMs[25]. While the operational principle of the Lawnmower is agnostic to the type of protease used in its design, trypsin is used here because of its retention of catalytic activity following a variety of chemical treatments including reduction, thiolation of amines, and subsequent covalent coupling to its thiols[25]. Microscale LMs were built around M-270 amine Dynabead hubs (diameter 2.8 μm; density 1.6 g/cm³; ThermoFisher 14307D). Reaction conditions were as previously described[25], except that a magnet was used to pellet beads during buffer exchanges and wash steps, and the incubation times were adjusted[51]. 20 mM MOPS pH 7.4 buffer is used throughout the synthesis. To synthesize microscale LMs, first, from an M-270 bead stock concentration of $2 \times 10^9$ beads/mL we pipetted out 10 μl which we washed and resuspended in MOPS buffer according to the manufacturers protocol for bead equilibration. 2 mg of the heterobifunctional crosslinker sulfo-SMCC (sulfosuccinimidyl-4-(N-maleimidomethyl) cyclohexane-1-carboxylate, ThermoFisher A39268) was dissolved in 200-300 μl of buffer. Beads were resuspended in this solution of sulfo-SMCC and incubated for 4 hours with rotary mixing at room temperature. 800 μl of buffer was added to this bead solution, beads were pelleted down and the supernatant removed. The beads were washed three times, each with 1000 μl of buffer to remove all unreacted sulfo-SMCC. In parallel, cysteines on trypsins (Sigma-Aldrich, T1426) were reduced for 60 minutes using TCEP gel (ThermoFisher 77712). 100 μl of 1 mg/ml cysteine-reduced trypsin was then added to the sulfo-SMCC beads, and these were incubated with rotary mixing for 4 hours at room temperature. After coupling, the total solution volume was brought to 1000 μl, the beads were pelleted, and the supernatant removed; this was repeated 5 additional times to ensure removal of all excess trypsin.

## Lawnmower polyvalency

To estimate the number of active trypsins per microsphere, we compared the rate of peptide cleavage for LMs and for varied concentration of trypsin in bulk solution. As a fluorogenic peptide was used (see "Lawn preparation" for details), we calculated the cleavage rate as the slope of increasing peptide fluorescence versus time (Supplementary Fig. 1E) in solutions where LMs were incubated with peptides for different times. Fluorescence was measured for solutions in silicone wells (Grace Bio-Labs) glued on a glass coverslip, using a Ti2-E inverted microscope (Nikon) with a Plan Apo λ 40x objective and LED illumination at 470 nm. We found that the cleavage rate for LMs at the concentration of $3 \cdot 10^7$ mL$^{-1}$ (counted in a Reichert Bright-Line Neubauer chamber) corresponds to the cleavage rate of free trypsin at $(2.6 \pm 0.6) \cdot 10^{-8}$ M (Supplementary Fig. 1F). Calculating the ratio between these, we obtained on average $(5 \pm 1) \cdot 10^5$ trypsins per LM, which corresponds to 0.02 trypsins per nm² of the surface area. This value represents the number of free trypsin equivalents: from previous work we learned that the activity of individual trypsins is unaffected by the chemical labeling performed here[25], but do not know how tethering to a large microsphere affects their activity.

## Lawn preparation

Substrate peptides were presented as a two-dimensional lawn on a glass coverslip, on which they were supported by an F127 (PEG-PPG-PEG) block-copolymer brush. The preparation of the "peptide" (peptide-F127) and "bare" (F127; Sigma P2443) lawns is as previously described and reproduced here[42]. The peptide lawn uses an NHS-functionalized F127 (Polymer Source P40768-EOPOEO2NHS) to couple the central lysine of the peptide (fluorescein isothiocyanate (FITC)-Ala-Pro-Ala-Lys-Phe-Phe-Arg-Leu-Lys-4-([4-(dimethylamino)phenyl]azo)benzoic acid (DABCYL), custom order from Biomatik) to the N-hydroxy-succinimidyl ester (NHS) moiety on the F127. The peptides provided as substrates in these LM experiments bridge a fluorophore and quencher, whereby peptide cleavage releases the quencher to solution and the surface-bound products become fluorescent[25,62]. In the current experiments, LM dynamics were tracked using bright-field microscopy, and fluorescence imaging was used only to confirm formation and accessibility of the peptide lawn (Ref. 42, Supplementary Figs. 13 and 14).

Glass was prepared for sample chambers as follows. Glass cover slips and slides were placed in 20 MΩ water and boiled for 10 minutes in a microwave. The glass was removed from water and submersed in acidified methanol (a mixture of equal volumes MeOH and HCl) and sonicated for 45 minutes. Acidified methanol was removed and the glass was rinsed with ddH₂O 5 times. The glass was then sonicated with ddH₂O for 5 minutes. Water was removed and then glass was sonicated in concentrated H₂SO₄ for 45 minutes. Glass was then rinsed with ddH₂O 5 times and then sonicated in ddH₂O for 10 minutes. Glass was then dried with airflow and baked at 100 °C for 10 minutes to remove remaining water. The slides were immediately silanized following this treatment. To do so, the glass was immersed in fresh Sigmacote (Sigma-Aldrich, SL2-100ML) for 1 minute 30 seconds and then baked for 30 minutes at 100 °C.

Glass chambers were constructed as previously described[42], using a glass slide, cover slip and double-sided tape as spacers. A 10% solution of NHS-F127 (or F127) in 1 M sodium phosphate buffer pH 6.0 was pipetted into each chamber to completely fill it. Chambers were immediately placed in a hydration chamber and incubated at 4 °C for 4 hours. The chambers were flushed with 1000 μl of the same buffer. A solution of 1 μl of peptide stock in DMSO with 59 μl of sodium phosphate buffer, pH 8.0 was created. Then 30 μl of peptide solution was pipetted into each chamber. Chambers were incubated overnight in a hydration chamber at 4 °C. Chambers were flushed with 0.1 M sodium phosphate buffer, pH 7.4 to remove excess peptide. Lawnmowers were then added in sodium phosphate buffer pH 7.4, at an appropriate dilution to provide well spaced trajectories in the field of view.

## Channel preparation

Microfabricated channels (Fig. 2) were designed to confine the LMs and provide quasi-1D peptide tracks along the channel floors. A difference in composition between the channel walls (hydrophilic) and channel floor (hydrophobic) was exploited to deposit the peptide lawn selectively on the hydrophobic channel floor. We achieved selective hydrophobicity on the channel floors by a combination of fabrication and surface modification. First, a thermally grown 100 nm thick SiO₂ substrate (⟨100⟩, Siegert Wafer GmbH) was used to form the floors of the channels. Ultrasonic rinsing of the substrates was carried out in acetone and isopropanol at room temperature (5 minutes in each). The substrates were further cleaned using oxygen plasma (Plasma Preen II-862, Plasmatic Systems, Inc., North Brunswick, NJ) to remove any excess organic debris. Subsequently, CSAR 62 (Allresist GmbH, Strausberg, Germany) was spin-coated at 2500 rpm to a 500 nm thickness and was subsequently baked at 180 °C for 2 min. Electron beam lithography (Voyager, Raith GmbH, Dortmund, Germany) was used to pattern microchannels on to the resist at an exposure dose of 250 μC/cm². Channels were developed by immersing the substrates in amyl acetate (Sigma-Aldrich, MO, USA) for 90 s, to dissolve the

exposed resist, and then rinsed in isopropanol (Sigma-Aldrich, MO, USA) for 30 s to remove the developer from the surface. Dimensions of channels (2.2 μm) were verified by atomic force microscopy (Bruker Icon 300, MA, USA). The developed microchannels were exposed to oxygen plasma (Plasma Preen II-862, Plasmatic Systems, Inc., North Brunswick, NJ) at 5 mbar for 30 s, which differentially affects the surface chemistry of the resist walls and the developed track floors[64,65]. Plasma-treated microchannel samples were surface modified using trimethylchlorosilane (TMCS) (Sigma-Aldrich, MO, USA) in vacuum-controlled chamber at 200 mbar, making the channel floors selectively hydrophobic[65] as needed for the F127 attachment[42]. Planar SiO$_2$ substrates without any resist were used as controls to verify the desired hydrophobicity after silanization by contact angle measurement ($90 \pm 2°$). The deposition of F127-peptide was confirmed by detecting the appearance of FITC fluorescence channels after addition of trypsin at 500 μg/mL in 0.1 M sodium phosphate buffer pH = 8 (Supplementary Figs. 13, 14).

### Imaging and tracking Lawnmower motion

Imaging of LMs was conducted using brightfield microscopy. For 2D experiments, experiments used a Zeiss Axioskop microscope with 10x objective and a FLIR Blackfly camera. Custom software was used to record image frames at 10-second intervals throughout Lawnmower experiments. Experiments on 2D peptide lawns were each run for a total of 12.5 hours, while experiments on 2D bare lawns were run for 6.2 hours. For channel experiments, imaging of LMs in channels relied on the light reflected from the Si substrate. A 40X Nikon objective and Photometics 95B camera were used in a basic custom-built upright microscope. Experiments in channels were run at 1- or 5-second intervals between frames, for 1.9 or 11.8 hours on peptide lawns, and for 2 hours for unmodified beads on bare lawns. In all 2D experiments, LM density was sufficiently low to maintain non-overlapping trajectories throughout the experiment, while the channel data includes partially overlapping trajectories. LM trajectories were determined from image stacks using the Fiji plugin MTrack2[66]. Trajectories in channels were analyzed for motion parallel and perpendicular to the channel axis. Analysis of channel data considered only portions of trajectories that remained within a given channel (Supplementary Fig. 15).

### Analytical methods

Displacement distributions of LMs on 2D bare lawns were fit with the Rayleigh distribution for 2D diffusion[67]:

$$P(\Delta r, \Delta t) = \frac{\Delta r}{2(D\Delta t + \sigma_t^2)} e^{-\frac{\Delta r^2}{4(D\Delta t + \sigma_t^2)}} \quad (1)$$

Here, $\Delta t$ is the time interval over which the displacements $\Delta r$ were determined, $D$ is the diffusion coefficient, and $\sigma_t^2$ is the experimental uncertainty in bead position tracking. To extract $D$ for each LM, its displacement distribution $P(\Delta r, \Delta t)$ is computed at increasing $\Delta t$ intervals, and the linear function $D\Delta t + \sigma_t^2$ found through fitting to Eq. 1 is used to determine its value of $D$ (Supplementary Fig. 8C).

LM interval speeds were calculated from overlapping displacements over each $\Delta t = 10$ second interval between image frames by

$$v_i = \frac{\Delta r_i}{\Delta t} = \frac{\sqrt{(x_{i+1} - x_i)^2 + (y_{i+1} - y_i)^2}}{\Delta t} \quad (2)$$

where $i$ is used to index the time intervals. The average speed of a LM throughout its trajectory was determined with

$$\bar{v} = \overline{\Delta r}/\Delta t = \frac{1}{n\Delta t} \sum_{i=0}^{n-1} \Delta r_i \quad (3)$$

where $n\Delta t = T_{msr}$, the measurement duration of the trajectory (12.5 hours for LMs on 2D peptide lawns and 6.25 hours for LMs on bare 2D lawns).

The mean squared displacement was calculated in two ways denoted by MSD$_{EA}$ and MSD$_{TA}$[59]. The ensemble-averaged MSD$_{EA}$ is computed as a function of absolute time $t$ for an ensemble of $N$ trajectories:

$$\text{MSD}_{EA}(t) \equiv \langle \Delta r^2(0,t) \rangle = \frac{1}{N} \sum_{j=1}^{N} \Delta r_j^2(0,t) \quad (4)$$

$\Delta r_j(0,t)$ is the displacement of the $j^{th}$ Lawnmower at time $t$, since the start of its trajectory at time $t = 0$. The trajectory-averaged MSD$_{TA}$ is computed independently for each trajectory using all displacements occurring for each time lag $\tau$:

$$\text{MSD}_{TA,j}(\tau) \equiv \overline{\Delta r_j^2(\tau)} = \frac{\Delta t}{T_{msr,j} - \tau + \Delta t} \sum_{t=0}^{T_{msr,j}-\tau} \Delta r_j^2(t,\tau) \quad (5)$$

$\Delta r_j(t,\tau)$ is the displacement of the $j^{th}$ Lawnmower at a time $\tau$ relative to its position at time $t$. The normalization prefactor is the inverse of the number of data points that contribute to the average for each $\tau$. $\Delta t = 10$ s is the time interval over which displacements are recorded in the experiment and $T_{msr,j}$ is the total measurement duration of the $j^{th}$ trajectory; both $T_{msr,j}$ and $\tau$ are integer multiples of $\Delta t$. To characterize the ensemble behavior of the LMs, the ensemble average of MSD$_{TA}$ is determined:

$$\langle \text{MSD}_{TA}(\tau) \rangle = \frac{1}{N} \sum_{j=1}^{N} \text{MSD}_{TA,j}(\tau) \quad (6)$$

Anomalous diffusion exponents $\alpha$ were determined by the slope of a linear fit of log MSD$_{EA}$ versus log time $t$ or of log MSD$_{TA}$ versus log time lag $\tau$. MSD$_{TA}$ and hence $\alpha_{TA}$ were determined to a maximum time lag of $\tau_{max} = 0.1T_{msr}$, where $T_{msr}$ is the total measurement duration. We selected $\tau_{max}$ of 0.1 to maintain reasonable statistics, as it has been shown that using larger lags (approaching the trajectory length) can result in significant variation of MSD$_{TA}$ for highly heterogeneous systems[68].

Directional persistence of LMs in 2D was assessed via the angular change between consecutive steps in the trajectory. Steps were converted from Cartesian to polar increments. If the trajectory is the set of points, $(x_i, y_i)$, then the polar increments are given by

$$(\Delta r_i, \Delta \theta_i) = \left( \sqrt{(x_{i+1} - x_i)^2 + (y_{i+1} - y_i)^2}, \tan^{-1} \frac{y_{i+1} - y_i}{x_{i+1} - x_i} \right) \quad (7)$$

The difference in angular increments, $Mod(\Delta\theta_{i+1} - \Delta\theta_i, 2\pi)$, provides an assessment of directionality: values near zero indicate a persistent step and values near $\pm \pi$ indicate an anti-persistent step.

Probability distributions of displacements along the channels $P(\Delta x)$ were compared with Gaussian distributions in two ways. Displacement distributions per one-second frame were extracted from trajectories in peptide lawn channels (Fig. 2C) and in bare lawn channels (Fig. 2D). The kurtosis of each probability distribution was determined:

$$\kappa = \frac{m_4}{\sigma^4} = \frac{\langle (X - \mu)^4 \rangle}{\langle (X - \mu)^2 \rangle^2} \quad (8)$$

Here, $m_4$ is the fourth moment, $\sigma$ the standard deviation, and $\mu$ the mean of the distribution of displacements X. A Gaussian distribution has $\kappa = 3$. Larger values of kurtosis correspond to distributions with

heavier tails. We also visualized the difference from a Gaussian distribution by taking the ratio of the measured $P(\Delta x)$ to a Gaussian distribution $P_G(\Delta x)$ with the same mean $\mu$ and standard deviation $\sigma$. This ratio highlights the tailedness of measured distributions. For the analyses presented in the main text, immotile periods for LMs in peptide lawn channels were disregarded and trajectories were truncated when LMs became immotile. We also explored the effects of including and excluding these qualitatively determined periods of immotility from the kurtosis analysis, and the effect of sampling at different time intervals (Supplementary Fig. 16 and Supplementary Note 3).

## 1D Lawnmower model

We modeled the LM by a random walker that is more likely to step towards previously unvisited positions when reaching the front or rear edge of the area it had already explored (Supplementary Fig. 10). In the region of bare lawn the LM is a usual random walker with probabilities $q$, $p$ to jump to the left/right, respectively, with $p + q = 1$. For an unbiased random walker, $p = q = \frac{1}{2}$; an overall net drift can be accounted for by setting $q = \frac{1}{2} - \delta$, $p = \frac{1}{2} + \delta$ ($-\frac{1}{2} \leq \delta \leq \frac{1}{2}$). At the edges of the bare region, stepping towards an uncleaved position is favored by the enzymatic action $a$ ($0 \leq a \leq \frac{1}{2}$), which adds to the probability of an outward step towards uncleaved lawn and decreases the probability of an inward step towards bare lawn (Supplementary Fig. 10). If the LM steps to an uncleaved position, this position is cleaved and becomes part of the bare region.

We scaled trajectories from this discrete, dimensionless model to compare with experiment. By choosing 20,000 steps in our model to correspond with one second in experiment, the step size of the random walker and the drift parameter $\delta$ were scaled to reproduce the experimentally observed diffusivity of the LM and (small) background flow, i.e., the variance $\sigma^2$ and mean displacement per 1-s frame. This provided a scaling of 2.35 nm per model step size.

For $a = 0$, the displacement distribution of the random walker is a Gaussian (kurtosis $\kappa = 3$) by construction, with first and second moments matching the experimental values. To account for the increased statistical weight of large displacements (larger than about $4\sigma$ of the Gaussian) observed in experiment on peptide channels, enzymatic action in the model was set to $a = 0.12$, which resulted in a kurtosis $\kappa > 3$ consistent with experiment. According to the model, these large displacements occur exclusively at the edges of the bare region, in contrast to what is observed in experiment (see Section "Statistical Analysis of LM displacements within previously visited regions"). We attribute this difference to the model assumption of perfect cleavage of the lawn at first contact with the LM, whereas the experimental LM appears to cleave only part of the lawn, leaving a mixture of cleaved and uncleaved lawn.

Enzymatic action, represented by $a = 0.12$ in the model, appears to amplify the effect of the drift near the edges, and therefore increases the average displacement per frame. This interplay of drift and enzymatic action biases LM action towards the direction of the background flow. Therefore, we needed to reduce the background drift correction $\delta$ by almost 40% to correctly reproduce the average displacement per frame observed in experiment ($\delta = 0.000057$ compared to the value of $\delta = 0.00009$ for $a = 0$).

## Comparison of experimental and model displacement distributions with Gaussian distributions

To determine whether LM displacements in channels can be described by normal diffusion, displacement distributions were compared with Gaussian distributions. Displacement distributions per one-second frame were extracted from trajectories in peptide lawn channels (Fig. 2C) and in bare lawn channels (Fig. 2D). Immotile periods for LMs in peptide lawn channels were disregarded: for this analysis, trajectories were truncated when LMs became immotile. The model trajectories were generated by numerical simulations as described above. The

resulting displacement distributions were characterized by the following statistical measures: peptide lawn channels: mean displacement 0.009 μm, standard deviation $\sigma = 0.33$ μm, kurtosis $\kappa = 3.3$; bare lawn channels: mean displacement 0.003 μm, $\sigma = 0.35$ μm, $\kappa = 3.0$; 1D LM model: mean displacement 0.009 μm, $\sigma = 0.33$ μm, $\kappa = 3.2$. Note that in all cases a displacement of about ±1.3 μm corresponds to $4\sigma$, such that the data for even larger displacements is dominated by statistical fluctuations and therefore has been omitted. A value of $\kappa > 3$ was found for these LMs in peptide channels at all timelags tested: $\tau = 1$ s, 5 s and 10 s.

## Statistical analysis of LM displacements within previously visited regions

The above simple model for a 1D LM suggests that LM displacements should be larger near the edges of a cleaved region, i.e., adjacent to uncleaved lawn where enzymatic action can take place. To find out whether the magnitude of LM displacements in the 1D channel is related to the position at which they occur relative to the border between previously visited and unvisited regions, we performed the following analysis.

We first corrected the whole ensemble of trajectories for the background flow, which was estimated from the mean value of all measured displacements. Given such a drift-corrected experimental trajectory $x(t_i)$ (with $x(t_0) = x(0) = 0$), where $t_i$ is the time point at which frame $i$ has been recorded, we iterated through all time points $t_i$ ($i > 0$) and kept track of the edges of the region already visited, i.e., we stored the minimal and maximal positions visited before $t_i$, $x_{max}(t_i) = \max_{j \leq i}(x(t_j))$ and $x_{min}(t_i) = \min_{j \leq i}(x(t_j))$. We scaled each position $x(t_i)$ such that the minimal and maximal positions correspond to ±1,

$$\tilde{x}(t_i) = \frac{2x(t_i) - (x_{min}(t_i) + x_{max}(t_i))}{x_{max}(t_i) - x_{min}(t_i)} \quad (9)$$

Then we subdivided the interval from −1 to +1 into 20 histogram bins to count how often a certain displacement $\Delta x(t_i) = x(t_i) - x(t_{i-1})$ occurred at the scaled position. The displacements were selected by their magnitude in three different ways: (i) we counted all displacements, (ii) we counted only displacements larger than $1\sigma$, and (iii) we counted only displacements larger than $2\sigma$; $\sigma$ is the standard deviation of the distribution of all measured displacements. When restricting to larger displacements, e.g. $3\sigma$, statistical fluctuations became dominant, because such large displacements are quite rare.

Even for a simple standard diffusion process (no LM action) the histogram obtained by counting all displacements was non-uniform, and thus hard to interpret. For this reason, we compared the experimental LM histograms to numerically generated histograms for standard diffusion (produced using the experimental diffusion coefficient). We performed this comparison by calculating the ratio between the probability of observing displacements within a specific histogram bin to the probability of observing displacements within the same histogram bin in the diffusion data (and selected by the same magnitude criterion). The results are shown in Supplementary Fig. 11. Values larger (smaller) than 1 indicate that displacements are observed more (less) frequently at this scaled position in the LM than they are observed in standard diffusive motion. The error bars estimate the statistical uncertainty, and are obtained from the standard deviation among 5 randomly chosen cluster samples of the full data set.

## Statistics and reproducibility

No statistical method was used to predetermine sample size. Sample sizes were determined by the number of particles in the field of view throughout each experiment. No single particles tracked throughout the experiment(s) duration were excluded from analysis. The experiments were not randomized. The Investigators were not blinded to allocation during experiments and outcome assessment.

## Reporting summary

Further information on research design is available in the Nature Portfolio Reporting Summary linked to this article.

## Data availability

The datasets generated during and/or analyzed during the current study are available in the accompanying source data files. Source data are provided with this paper.

## Code availability

Codes written to perform the analyses presented in this manuscript are available on GitHub (https://github.com/chapSKor/lawnmower).

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

## Acknowledgements

This project was spawned many years ago by a team grant from the Human Frontier Science Program. From this original group, we would particularly like to acknowledge Laleh Samii and especially Suzana Kovacic for their work on the original Lawnmower design, Martin Zuckermann for physical insight, Dek Woolfson for stimulating ideas, and Damiano Verardo for many discussions and complementary experiments in Lund. We thank our former Lund colleague Jingyuan Zhu for inspiring discussions of channel fabrication. The channels were made in Lund Nano Lab. We are grateful to present and former SFU colleagues Mike Kirkness for help with surface chemistry and critical questions, Juliette Savoye for testing of experimental protocols, David Lee for microscopy assistance, and Eldon Emberly and David Sivak for insightful theoretical discussions. Nordita is partially supported by Nordforsk. This research has been supported by Natural Sciences and Engineering Research Council of Canada (NSERC) Discovery Grants RGPIN-2020-04680 and RGPIN-2015-05545 (NRF), NSERC Post-Graduate Scholarship – Doctoral and Postdoctoral Fellowship (CSK), Swedish Research Council projects 2020-04226 (HL) and 2020-05266 (RE), Nordforsk (RE), European Union's Horizon 2020 research and innovation programme under grant agreements No 732482 (Bio4Comp) and no 951375 (ArtMotor) (HL, PMGC) and Volkswagen Foundation Project No. 93439/93440 (HL, RL).

## Author contributions

Conceptualization: CSK, IU, PS, RL, HL, NRF. Formal Analysis: CSK, IU, RE, CA. Methodology: CSK, IU, PS, RL, RE, HL, NRF. Investigation: CSK, IU, PS, RL, RE. Funding acquisition: RE, HL, NRF. Supervision: HL, NRF. Project administration: NRF. Visualization: CSK. Writing – original draft: CSK, IU, NRF. Writing – review & editing: CSK, IU, PS, RL, RE, CA, PMGC, HL, NRF.

## Competing interests

The authors declare no competing interests.
