## [Peer review file · Nature Communications]

REVIEWER COMMENTS

Reviewer #1 (Remarks to the Author):

In this manuscript, authors demonstrate a trypsin-based artificial molecular motor, termed as the Lawnmower (LM). The LM operates as a burnt-bridge Brownian motor on the surface of a substrate peptide lawn. The authors also demonstrate biased 1D motion of the LM on the microfabricated linear tracks. I agree with the authors' notion that this is the first demonstration of an autonomous artificial molecular motor based on a protein (an enzyme) and a substrate peptide, although DNA/RNA-based artificial molecular walkers/motors also use DNA/RNA cleaving enzymes such as DNAzyme (Ref 8) and RNase H (Ref 15). On the other hand, the materials and methods are not described in sufficient detail, the mechanism of long immotile state is not clear, and the validity of the analysis of anomalous motion is not convincing. Furthermore, performance of the LM is not superior to that of the relevant DNA/RNA-based motor previously reported (Ref 15) in terms of the unidirectionality and velocity. Therefore, I have a reservation about the publication of the present manuscript in Nature Communications.

Major points

1. It is difficult for me to understand the details and validity of the experiments, especially the preparation and biochemical characterization of the LM, because I have no access to Supporting Ref 1 (an IEEE Transactions). Supporting Ref 2 (a doctoral thesis?) is also difficult to access. In my understanding, manuscripts published in Nature Communications should include details of materials and methods, rather than citing previous studies and stating "as previously described." A number of important pieces of information are missing in the present manuscript, such as the source of trypsin enzyme used.
2. What is the multivalency of the LM used in the present study? Although the authors describe that "Kovacic et al. determined that the multivalency of the QD LM is only $N \approx 8$ trypsins (1)" (Supporting Information, page 2 line 9) and "Microscale LMs have the potential for thousands of trypsins to be simultaneously engaged with peptides on the lawn" (Main manuscript, page 5 line 26), they do not quantitatively describe the multivalency of the LM.
3. The mechanism of long immotile state lasting for more than 1 hour is not clear for me (Fig. 1C and Fig. S2), although the authors describe a possible mechanism as "When LM-track interactions are sufficiently high in number, the LM may become immotile, as its diffusion is quenched by the polyvalency of binding interactions (50)" (Main manuscript, page 5 line 27). If this is the case, how does the LM escape from the long immotile state and why does it take so long time before escaping? I anticipate that the LM with high valency can easily escape from the immotile state if all trypsin molecules on the bead surface are catalytically active. However, the authors do not describe the activity of the trypsin used in the present study quantitatively, but just describe as "Trypsin activity is retained throughout this treatment (1)" (Supporting Information, page 2 line 22).

4. To understand the mechanism of the long immotile state, it is highly recommended to check the 2D trajectories of the LM carefully to confirm if the motions are really self-avoiding and the “entrapments” occur at the locations where the surrounding tracks are already visited (cleaved). In the relevant study of a multivalent DNA-microparticle artificial motor (Ref 15), it has been clearly shown that entrapments lasting for tens of minutes occur at the locations where the surrounding RNA lawns are cleaved.

5. In my understanding, the analysis of ensemble-averaged MSD (MSDEA) and anomalous factor α (α_{EA}) is conducted after the separation of motile and immotile states (Fig. 1D). On the other hand, trajectory-averaged MSD (MSDTA) and anomalous factor α (α_{TA}) are calculated by using entire trajectory of each LM. For quantitative comparison, the motile and immotile states should be separated also for the trajectory-averaged analysis.

6. I do not understand the description about the Simple 1D LM model, “we chose 20000 random steps of size 0.00235 μm to correspond to a one-second frame in experiment; this relatively large number of steps per frame entails a spatial resolution comparable to the experimental one” (Supplementary Information, page 5 line 11), because “spatial resolution” of optical microscopy will be several hundred nanometers and much larger than 0.00235 μm .

7. I am not fully convinced of the necessity and validity of the section “Motor-like motility does not require $\alpha > 1$ ” to interpret the experimental value of α_{TA} close to 1 (Supporting Information, page 7 line 24). For the motile and immotile states, I anticipate the $\alpha > 1$ and $\alpha < 1$, respectively, and the mixture of two states would result in $\alpha \sim 1$. Again, I think that the motile and immotile states need be separately analyzed to understand the motional behavior of the LM. Furthermore, the rationale for employing a Mittag-Leffler distribution to calculate MSD is not clear for me (Fig. S16).

Minor points

1. Fig. 1B. It is difficult to see the details of individual trajectories. It is recommended that each trajectory be shown separately and not superimposed. Also, the scale of the horizontal axis in the inset seems to be incorrect (not “100” but “50” for right side?).

2. Why does a background flow of $\sim 0.01 \mu\text{m/s}$ occur? (Main manuscript, page 4 line 43). This value is by no means “small” compared to the velocity of molecular motors. The background flow must be kept as low as possible.

3. Ref 36 is not appropriate because this paper describes the mechanical property of the cellulose at the nanoscale, not cellulase as a burnt-bridge molecular motor. I suggest to cite following papers as landmark studies. K. Igarashi, et al., J. Biol. Chem 284, 36186-36190 (2009) and S. K. Brady, et al., Nat. Commun. 6, 10149 (2015).

4. Supporting Information, page 2 line 8. Related to the description of the processivity, artificial molecular motors based on the micron-sized bead will not diffuse away from the track surface because the mass is large and effect of the gravity is not negligible. Therefore, in contrast to the nano-sized molecular motors, its apparent processivity will become very high even if the multivalency is low. I think this point needs to be clarified.

5. Supporting Information, page 3 lines 31. Not “50 $\mu\text{g}/\text{mL}$ ” but “50 or 500 $\mu\text{g}/\text{mL}$ ”. Description is not consistent with Fig. S14.
6. Supporting Information, page 7 lines 13 and 18. Not “Eq. S6” but “Eq. S8”.
7. Supporting Information, page 7 line 16. Not “Eq. S7” but “Eq. S9”.
8. Supporting Information, page 7 line 33. Not “Figure S15” but “Figure S16”.
9. Supporting Information, page 21, line 4. Not “Si” but “SiO₂”.
10. Fig. S2 and Fig. S3. It would be very helpful to readers if the horizontal axes are shown in time (h) instead of frame number, as Fig. 1C.
11. Fig. S12. The label and unit in the vertical axis are missing.
12. Fig. S16. What are the units of horizontal (Time) and vertical (MSD) axes? Please clarify.
13. In the Movies S1 and S2, I found several dust-like contaminations. I wonder what these contaminations are and how these contaminations affect the LM motion. I would appreciate the comments from the authors.

Reviewer #2 (Remarks to the Author):

Korosec et al. describe a completely protein-based synthetic motor that operates under a burnt-bridge Brownian ratchet (BBR) mechanism. Specifically, they show that trypsin functionalized microspheres have directional diffusion on a peptide substrate lawn or 1D track. These results are a significant advancement in the field of synthetic motors because the authors use protein building blocks instead of DNA, which inherently have more diverse functions than DNA. Though the motor does not move in continuous motion, but rather moves in bursts and is frequently ‘stalled’, this is a first iteration and conceptually is very important to the field. However, there is much improvement that needs to be made to the manuscript to better communicate the findings to a broad audience, such as the readership of Nature Communications. The text is bare bones, oftentimes leaving the reader to have to interpret what experiment/analysis is being performed directly from the Figures rather than being explained in the text. In addition, some of the claims need to be toned down. Lastly, very little detail is included characterizing the motor and track synthesis.

Specific Comments:

- Page 2, line 8: “Motion of a synthetic motor along a pre-defined 8 track, a prerequisite for many potential nanotechnological applications, has to our knowledge not yet been realized.”

Many of the motors cited throughout the paper operate along a pre-defined track.

- Page 2, line 14: “but to our knowledge, motility of a synthetic motor constructed of proteins – the 14 material system that enables the complexity of life – has yet to be demonstrated”

The authors cite a paper in the supplemental describing such a motor. Should reword to say is an ‘important ongoing challenge’ to be more accurate. In addition, the authors should add this discussion in the main text, comparing and contrasting the motors, and properties.

- Page 2, line 29: “Many 29 different biological systems undergo directed motion as BBRs (28-33), the mechanism of choice 30 for regulated enzymatic degradation of key extracellular scaffolds in animals and plants (34-38).”

List some of these biological systems, rather than making the reader look up these sources.

- The details regarding motor synthesis and track synthesis and characterization are severely lacking. For the motor synthesis, the authors simply cite Figure S1, which has the wrong chemical structure for the maleimide. For example, how many motors are attached. Is the system even multivalent. Also, there are very little to no details regarding the surface functionalization. Based on the methods, the authors are using a fluorescent substrate, but this is never described in the text, nor is this used to characterize the surface. Rather, there is a single figure (Fig S14) in the supplemental showing a fluorescent image of ‘cleaved tracks’, yet this is never cited in the main text. Also, figures S13-S16 are not cited in the text.
- How does immobilization affect trypsin activity? Can you please compare k_{cat} on surface using soluble substrate compared to k_{cat} free in solution? Also, please add a little more discussion about trypsin. There is a great body of literature describing its processive activity.
- Page 2, line 44 in supplemental: The authors mention in the SI that the surface functionalization ‘The F127 brush of the bare lawn did an outstanding job of blocking nonspecific adhesion of the LMs to the surface (<1% immotile), consistent with its previously demonstrated blocking abilities with microspheres of other surface chemistries’, yet the results show that these particles move much less on these surface compared to peptide surfaces. Can the authors please elaborate more on this difference, and add to main text when describing the motor and surface.
- Please be more quantitative rather than descriptive when discussing the results. For example, ‘Over similar timescales LMs on 2 peptide lawns travel much farther than on bare lawns.’ How much further? Instead, describe results as, the motors reached dR of 6 μ m, where for bare lawns, had dR of 0.6 μ m.
- Page 3, line 10: ‘A threshold value’ - What is the value? How was this value determined? Please explain in text.
- What is time interval for Figure 1C? It is mentioned in supplemental, but not main text.
- Fig. S5, why is one motor superdiffusive?
- Why does MSD-TA versus MSD-EA vary significantly. Can you explain why you switched analysis in text.
- Figure 1E: What exactly is $P(dr)$. Please define/describe in text. This is an example where the reader has to interpret what analysis is being performed. Also, please just show Inset for figure 1E. Can move the ‘combined’ as a supplemental figure.

- Figure 1F: Why is there a larger distribution showing high speeds, when motion is described mostly as saltatory? Also, the authors mention that the motor is oftentimes stalled, which underestimates alpha, so why is the largest bin the fastest speeds. This doesn't agree with Fig. 1c.

- Please describe Figure 2b in more detail. There looks like a very interesting result in bottom picture, fourth path from the right. Looks like there are two particles in same track, where the one particle runs into 'cleaved' substrate (i.e., product) and stops. What if you washed the surface and added particles again, would see a larger number of particles that stop because encountering 'cleaved' substrate? Also, what if you start the particles in a direction (due to gravity), then observed motion. Do all the particles move in the same direction. To do this experiment, the authors could make a 'binding buffer' with edta, to inactivate trypsin, then wash the surface with 'rxn buffer'? Lastly, Figure 2B shows same trajectories in S15, but Figure S15 has strange spots throughout? What are these spots? Can also see spots in Figure 2B, but less pronounced.

- Should show Fig. 2c and 2d in same length scale.

- What exactly is Fig. 2E? Should explain more in main text. Again, experiment and analysis is not clearly explained in text to non-expert.

- Page 5, line 26: 'Microscale LMs have the potential for thousands of trypsins 26 to be simultaneously engaged with peptides on the lawn.'

What if you vary trypsin density? How does this affect velocity/diffusion? Along the same lines, multivalency has been shown to improve processivity. Since your motors seem to have non-specific interactions with the surface (based on observing little motion on bare surfaces), it seems that multivalency isn't needed? This would be a good section to compare to previous QD trypsin motor. I think this discussion and analysis is very important based on the body of literature of synthetic motors.

- A very important feature for BBM is that the motor has stronger affinity towards product compared to substrate. What exactly is this difference in affinity? Could you synthesize an 'all product' surface, and compare Kd's. With DNA-based motors, this can easily be measured and compared.

- Add discussion comparing and contrasting motors attached to particle versus free in solution. For example, once protein is denatured, the motor could get stuck if immobilized on a particle. Where having added in solution has a continuous supply of motor.

- Page 5, line 5. What exactly is kurtosis? Please explain in more detail for the reader.

For example, the authors just mention in text, and do not explain what this is or relate it to Fig. 2E.

'these LM displacement distributions exhibit a kurtosis $\kappa > 3$, indicating larger 5 displacements than expected from a normal distribution (Fig. 2E). In contrast, particles on bare 6 lawn have $\kappa \approx 3$ with displacements described by the expected Gaussian distribution for one-7 dimensional diffusive motion.'

- Please elaborate on S11 analysis. I don't really see a difference.

- Page 5, line 34. 'A heavy tail to the distribution of escape times from such a locally bound state can give rise to an MSD that scales sub-linearly with time, even for completely directional motion (Supplementary Text).'

Wouldn't a solution be to look at larger times to tease out superdiffusion, greater than the 'stuck time'. By showing MSD vs time at various intervals, you should be able to observe the phase in the curve where directional diffusion is observed. For example, this is what was done for Fig. 1d green. So, fit that data, and report what range was analyzed in the text.

- What affects do salt and temperature have on motor properties? A potential major advantage of protein motors compared to DNA motors is minimal dependence on salt and temperature, comparatively. The authors should investigate this.

We thank both reviewers for their careful reading and critical assessments of the submitted manuscript. We agree with many of their criticisms and have made significant changes, all indicated in red in the marked-up copies of the manuscript and supporting information. We feel that the manuscript is significantly strengthened thanks to their suggestions and questions. The reviewers highlighted the novelty, conceptual importance and significant advancement provided by this work in their reviews. We hope that our edits address their concerns regarding the details of the work and questions about the mechanism of LM action, and that the work will now be viewed as highly appropriate for *Nature Communications*.

Reviewer #1

In this manuscript, authors demonstrate a trypsin-based artificial molecular motor, termed as the Lawnmower (LM). The LM operates as a burnt-bridge Brownian motor on the surface of a substrate peptide lawn. The authors also demonstrate biased 1D motion of the LM on the microfabricated linear tracks. I agree with the authors' notion that this is the first demonstration of an autonomous artificial molecular motor based on a protein (an enzyme) and a substrate peptide, although DNA/RNA-based artificial molecular walkers/motors also use DNA/RNA cleaving enzymes such as DNAzyme (Ref 8) and RNase H (Ref 15). On the other hand, the materials and methods are not described in sufficient detail, the mechanism of long immobile state is not clear, and the validity of the analysis of anomalous motion is not convincing. Furthermore, performance of the LM is not superior to that of the relevant DNA/RNA-based motor previously reported (Ref 15) in terms of the unidirectionality and velocity. Therefore, I have a reservation about the publication of the present manuscript in *Nature Communications*.

We thank the reviewer for their very careful reading of our manuscript, and for their insightful comments and constructive criticisms of our work. We are glad that they support our claim for the first protein-based motor; there are certainly other motors that include protein or enzyme components, which we reference and discuss in this manuscript. However, to our knowledge, all of the existing synthetic motors made of biomolecular building blocks include nucleic acids as a core part of their design; the Lawnmower circumvents the need for nucleic acids. We note that these nucleic acid-based designs (DNAzyme-based: refs 6-8; RNase H-based highly polyvalent DNA motors (HPDMs): refs 9, 10, 13, 15) have seen multiple iterations in their performance, which led to a series of high-profile publications and stimulated significant theoretical modelling. Please note that we do not claim that the Lawnmower is superior to the performance of these other designs, but it does present a unique approach to molecular motor engineering, which exploits Nature's building block of choice for these systems: proteins. Furthermore, although we do not claim superiority, we would like to note – as we do in the manuscript – that the average LM speeds are very similar to those achieved by the HPDMs. The LM exhibits long stretches of near-ballistic motion with very high velocities (surpassing instantaneous velocities of HPDMs), interspersed with periods of quasi-immotility.

Shown in more detail below, we have added significantly to the manuscript in order to address the other points raised by the referee:

- We have rewritten, and moved to the main text, the methods section to remove the need to reference protocols in other documents.
- We have significantly strengthened the analysis and discussion of potential mechanisms underlying the immotile state, placing this better in context of other work in the literature.
- We have revised our discussion of anomalous diffusion and hope that this is now much clearer.

We believe we have addressed each of the reviewer's explicit points below, and hope that these responses – along with the substantial changes to the manuscript – leave the reviewer with no further reservations about the suitability of our work for *Nature Communications*.

Major points

R1.1. It is difficult for me to understand the details and validity of the experiments, especially the preparation and biochemical characterization of the LM, because I have no access to Supporting Ref 1 (an IEEE Transactions). Supporting Ref 2 (a doctoral thesis?) is also difficult to access. In my understanding, manuscripts published in Nature Communications should include details of materials and methods, rather than citing previous studies and stating "as previously described." A number of important pieces of information are missing in the present manuscript, such as the source of trypsin enzyme used.

Actions taken: We have extended the methods section for LM and surface synthesis extensively, so that the experiments can be reproduced without the need to refer to other sources. A full description of the methods is found in the main manuscript (moved from the supporting information). The sources of reagents - including trypsin - are provided throughout the methods section.

R1.2. What is the multivalency of the LM used in the present study? Although the authors describe that "Kovacic et al. determined that the multivalency of the QD LM is only $N \approx 8$ trypsins (1)" (Supporting Information, page 2 line 9) and "Microscale LMs have the potential for thousands of trypsins to be simultaneously engaged with peptides on the lawn" (Main manuscript, page 5 line 26), they do not quantitatively describe the multivalency of the LM.

Actions taken: We have performed additional experiments to answer this question, and obtained a value of 5×10^5 trypsins / LM for the microbead-based LMs used in this work (note that these are much larger than the QD LM discussed in Kovacic *et al.* (ref. 24). The new experiments are described in the Methods, and the polyvalency is now addressed in the main text:

"We verified the activity of LMs in solution and compared their peptide cleavage rate to that of trypsin (Fig. S1). We estimate that LMs each accommodate $(5 \pm 1) \cdot 10^5$ active trypsins, or about 0.02 trypsins per nm^2 of the surface area of the microsphere. Of these, ~2000 trypsins can be engaged with the underlying peptide lawn, which presents $\sim 10^4$ peptides within this LM footprint (Supporting Text). Thus, LMs are active and highly polyvalent, with the potential for thousands of interactions with the lawn."

We have also added a section to the supporting text (“Polyvalency Estimates”) that outlines these calculations.

R1.3. The mechanism of long immotile state lasting for more than 1 hour is not clear for me (Fig. 1C and Fig. S2), although the authors describe a possible mechanism as “When LM-track interactions are sufficiently high in number, the LM may become immotile, as its diffusion is quenched by the polyvalency of binding interactions (50)” (Main manuscript, page 5 line 27). If this is the case, how does the LM escape from the long immotile state and why does it take so long time before escaping? I anticipate that the LM with high valency can easily escape from the immotile state if all trypsin molecules on the bead surface are catalytically active. However, the authors do not describe the activity of the trypsin used in the present study quantitatively, but just describe as “Trypsin activity is retained throughout this treatment (1)” (Supporting Information, page 2 line 22).

The immotile state is seen also in other burnt-bridge motors such as the highly polyvalent DNA motors (HPDMs) and the Par system, cited within the text. There is no established mechanism for this state, but we now present and discuss two possible reasons in the text.

Action taken: Text has been added to present and describe possible mechanisms for the (lengthy) periods of immotility:

“Immotile dwells have been observed for the highly polyvalent DNA motors (HPDMs), where immotility was attributed to entrapment within previously visited, product-rich regions (15). We also see evidence of immotility coinciding with previously visited regions: for example, the long dwell from 4-6.5 hours indicated by the red colour in Fig. 1C overlays with the earlier portion of the trajectory indicated in blue (Fig. 1C, inset). Even small-length fluctuations would be sufficient to drive the LM into a previously visited region, which has the potential to lead to stalling due to the extremely high polyvalency of (even very weak) trypsin-product interactions (51, 52). A second mechanism for transient immotility in highly polyvalent systems is the long timescale associated with cleaving the large number of substrates under the footprint of the motor. In nanoscale HPDMs, release from such a state was posited to give rise to large displacements, associated with a heavy tail in the displacement distributions (10). This mechanism was demonstrated in simulations of the reconstituted microscale Par BBRs, which found step size to inversely correlate with the polyvalency of bead-track interactions during trajectories (43). It is likely the same dynamics are at play in LMs on peptide tracks, producing the observed saltatory dynamics and heavy-tailed displacement distributions (Figs. 1E,F; 2E,F).

Thus, although an increase in LM valency may lead to an increase in speed (10, 53), when LM-track interactions are sufficiently high in number, the LM has a higher probability of becoming immotile, as its motion is quenched by the polyvalency of binding interactions (52). Once enough bonds have broken, the LM is able to capture thermal fluctuations and rapidly access nearby peptides, and if cleavage is sufficiently rapid

compared to the formation of many new binding interactions, then large-scale motion continues until enough interactions stochastically form to again trap the LM locally in an immotile state (43). Such dynamics appear to be shared among emerging examples of BBRs, and warrant further investigations to optimize performance.”

Regarding the comment “I anticipate that the LM with high valency can easily escape from the immotile state if all trypsin molecules on the bead surface are catalytically active.” Our intuition is the opposite, since we consider the additional possibility of trypsin binding (albeit weakly) to products. While this is not as common a mechanism for trypsin as in other proteases (e.g. ref. 51), it is feasible. In our previously published simulation work on multivalent walkers (ref 52; no free-energy gradient or substrate cleavage mechanism was present), we derived an expression for their diffusion coefficient and found $D \propto 1/(\text{number of binders})^2$. Thus, here we hypothesize that an immotile state can result when the LM finds itself in a product-rich region. This is a similar mechanism proposed by Yehl et al. (ref. 15) to describe the transient entrapped state observed in their measurements of HPDMs. In this state the LM can be approximated as a multivalent diffusive walker with a very high valency. Based on our previous model results, and our new measurements that estimate up to 10^3 potentially engaged trypsins (response R1.2), this would result in $D \rightarrow 0$. Escape from this state requires (re)engagement with a substrate-rich region of the track. In order to move far enough to capture substrate in a new region, the vast majority of (weak) trypsin-product interactions must simultaneously unbind. This becomes increasingly unlikely as the number of binders/trypsins increases, leading to long dwell times in the immotile state.

Action taken: The new text – included above – clarifies the role of polyvalency.

To address the reviewer’s comment, “However, the authors do not describe the activity of the trypsin used in the present study quantitatively”, we have conducted further experiments to determine trypsin activity while bound to the beads. See response to question R1.2 above. The value of trypsin polyvalency of $(5 \pm 1) \cdot 10^5$ trypsins represents an equivalent of active trypsins on the bead, since the calibration curve used free trypsin in solution.

R1.4. To understand the mechanism of the long immotile state, it is highly recommended to check the 2D trajectories of the LM carefully to confirm if the motions are really self-avoiding and the “entrapments” occur at the locations where the surrounding tracks are already visited (cleaved). In the relevant study of a multivalent DNA-microparticle artificial motor (Ref 15), it has been clearly shown that entrapments lasting for tens of minutes occur at the locations where the surrounding RNA lawns are cleaved.

We have visually examined 2D trajectories for entrapment. One example of this can be seen in Fig. 1C, where a lengthy dwell from 4-6.5 hours is shown in red in the inset trajectory. This falls in the region previously explored by the LM (indicated by blue).

Action taken: We have added text to describe this evidence of entrapment:

“Immotile dwells have been observed for the highly polyvalent DNA motors (HPDMs), where immotility was attributed to entrapment within previously visited, product-rich regions (15). We also see evidence of immotility coinciding with previously visited regions: for example, the long dwell from 4-6.5 hours indicated by the red colour in Fig. 1C overlays with the earlier portion of the trajectory indicated in blue (Fig. 1C, inset). Even small-length fluctuations can be sufficient to drive the LM into a previously visited region, which has the potential to lead to stalling...”

R1.5. In my understanding, the analysis of ensemble-averaged MSD (MSDEA) and anomalous factor α (α EA) is conducted after the separation of motile and immotile states (Fig. 1D). On the other hand, trajectory-averaged MSD (MSDTA) and anomalous factor α (α TA) are calculated by using entire trajectory of each LM. For quantitative comparison, the motile and immotile states should be separated also for the trajectory-averaged analysis.

In our analysis, each *entire* trajectory was classified as either “motile” or “immotile” based on the extent of motion over the full 12.5-hour measurement duration (Fig. S4). The complete trajectories classified as “motile” were used to determine both MSD_EA and MSD_TA (and \langle MSD_TA \rangle). Thus, the same trajectories are used for both types of analysis. Segmentation into motile / immotile dwells is not performed within trajectories.

Action taken: The text has been revised to make both the classification and the subsequent analysis clearer:

“In fact, while the majority of LMs exhibited motile dynamics on peptide lawns (55%; number of LMs $n = 59$), a significant fraction remained immotile throughout the 12.5-hour experiment ($n = 49$). We classified a LM as “motile” if its mean-squared displacement exceeded a threshold value of $10 \mu\text{m}^2$ at $\tau = 4400 \text{ s}$, and “immotile” otherwise (Fig. S4A). To correct for sample drift, trajectories of immotile LMs were averaged and this average trajectory was subtracted from each motile LM trajectory (Fig. S4B). Our characterization of LM dynamics focuses on these drift-corrected trajectories of the motile class of LMs: while they may contain transient immotile dwells, they all contain active periods of motility, and the entire trajectories were analysed without internal segregation. Later, we discuss possible reasons for immotility, observed only on peptide lawns.”

R1.6. I do not understand the description about the Simple 1D LM model, “we chose 20000 random steps of size $0.00235 \mu\text{m}$ to correspond to a one-second frame in experiment; this relatively large number of steps per frame entails a spatial resolution comparable to the experimental one” (Supplementary Information, page 5 line 11), because “spatial resolution” of optical microscopy will be several hundred nanometers and much larger than $0.00235 \mu\text{m}$.

The model is initially dimensionless and the time and length scales are then scaled to provide a match to experimental trajectories. We agree that this description was not clear.

Action taken: We have rephrased the text (now in the Methods section of the main manuscript) to read

“We scaled trajectories from this discrete, dimensionless model to compare with experiment. By choosing 20,000 steps in our model to correspond with one second in experiment, the step size of the random walker and the drift parameter δ were scaled to reproduce the experimentally observed diffusivity of the LM and (small) background flow, i.e., the variance σ^2 and mean displacement per 1-s frame. This provided a scaling of 2.35 nm per model step size.”

R1.7. I am not fully convinced of the necessity and validity of the section “Motor-like motility does not require $\alpha > 1$ ” to interpret the experimental value of α_{TA} close to 1 (Supporting Information, page 7 line 24). For the motile and immotile states, I anticipate the $\alpha > 1$ and $\alpha < 1$, respectively, and the mixture of two states would result in $\alpha \sim 1$. Again, I think that the motile and immotile states need be separately analyzed to understand the motional behavior of the LM. Furthermore, the rationale for employing a Mittag-Leffler distribution to calculate MSD is not clear for me (Fig. S16).

The referee raises a valid concern that we have considered very carefully. We have attempted to perform within-trajectory segregation of states into motile and immotile, but the objective assignment of transient states proved very challenging in the context of the significant heterogeneity of dynamics within individual trajectories. We concluded that such separate analysis would be unreliable and raise questions about the details of the segregation.

For these reasons our state-dependent analysis focuses on classifying entire trajectories as motile or immotile, followed by performing MSD analysis separately on each class (e.g. Fig. 1D,F, S4). We note explicitly in the manuscript the heterogeneity of dynamics and its effects on the average values obtained through the analysis:

“LMs exhibit extended immotile dwells during their trajectories (Fig. 1C) that decrease their average speed \bar{v} ”

“We expect that $\langle \alpha_{TA} \rangle = 1.1$ underestimates the superdiffusivity of active LMs because this treatment assumes time-invariant dynamics and averages out the bursts of active motion with the dwells of immotility on peptide lawns (44).”

Regarding the mentioned section in the Supporting Information, this perspective provided us a distinct approach to interpreting MSDs, but we agree that it is not necessary for the current manuscript as it is purely speculative and thus not essential.

Action taken: we have removed this section from the Supporting Information and the accompanying discussion from the main text.

Minor points

R1.8. Fig. 1B. It is difficult to see the details of individual trajectories. It is recommended that each trajectory be shown separately and not superimposed. Also, the scale of the horizontal axis in the inset seems to be incorrect (not “100” but “50” for right side?).

The displacements (Δr) versus time for each of the trajectories in Fig. 1B are plotted individually in separate panels in Fig. S2.

The scale in the horizontal axis of the inset has been corrected – thank you for catching this!

R1.9. Why does a background flow of $\sim 0.01 \mu\text{m/s}$ occur? (Main manuscript, page 4 line 43). This value is by no means "small" compared to the velocity of molecular motors. The background flow must be kept as low as possible.

We agree about the desire to keep background flow as low as possible. In our experimental design, particles are loaded from one side of the chamber. It is possible that some residual flow remains following the addition of particles. In the channel experiments, the value of $\sim 0.01 \mu\text{m/s}$ for residual flow was obtained by comparison to simulations. In the 2D LM experiments, the average drift was significantly lower: over 12 hours of experiments, the average downward displacement was $9 \mu\text{m}$, leading to a negligible drift of $\sim 0.0002 \mu\text{m/s}$ (0.2 nm/s).

R1.10. Ref 36 is not appropriate because this paper describes the mechanical property of the cellulose at the nanoscale, not cellulase as a burnt-bridge molecular motor. I suggest to cite following papers as landmark studies. K. Igarashi, et al., J. Biol. Chem 284, 36186-36190 (2009) and S. K. Brady, et al., Nat. Commun. 6, 10149 (2015).

We agree and thank the reviewer for these suggestions.

Action taken: we have updated the citations accordingly.

R1.11. Supporting Information, page 2 line 8. Related to the description of the processivity, artificial molecular motors based on the micron-sized bead will not diffuse away from the track surface because the mass is large and effect of the gravity is not negligible. Therefore, in contrast to the nano-sized molecular motors, its apparent processivity will become very high even if the multivalency is low. I think this point needs to be clarified.

Action taken: In expanding the methods section and moving it to the main text, we removed this discussion. The effects of gravity on the LM are discussed in the manuscript on p. 4 (2D) and on p. 5 (channels):

“Gravity holds the LMs against the lawn; the observation of 2D diffusive motion is consistent with the previously demonstrated ability of the F127 surface to block nonspecific adhesion of these microspheres (41).”

“Gravity constrained the dense beads within these $0.5 \mu\text{m}$ deep channels, which provided effective 1D confinement.”

R1.12. Supporting Information, page 3 lines 31. Not “ $50 \mu\text{g/mL}$ ” but “50 or $500 \mu\text{g/mL}$ ”. Description is not consistent with Fig. S14.

We are grateful to the reviewer for the careful reading that found this and the following inconsistencies.

Action taken: 50 has been amended to 500 $\mu\text{g/mL}$ in the methods.

R1.13. Supporting Information, page 7 lines 13 and 18. Not “Eq. S6” but “Eq. S8”.

Action taken: Equation numbering has been corrected (and moved to the methods within the main text).

R1.14. Supporting Information, page 7 line 16. Not “Eq. S7” but “Eq. S9”.

Action taken: Equation numbering has been corrected (and moved to the methods within the main text).

R1.15. Supporting Information, page 7 line 33. Not “Figure S15” but “Figure S16”.

Action taken: Equation numbering has been corrected (and moved to the methods within the main text).

R1.16. Supporting Information, page 21, line 4. Not “Si” but “SiO₂”.

Action taken: Equation numbering has been corrected (and moved to the methods within the main text).

R1.17. Fig. S2 and Fig. S3. It would be very helpful to readers if the horizontal axes are shown in time (h) instead of frame number, as Fig. 1C.

Action taken: These labels have been updated.

R1.18. Fig. S12. The label and unit in the vertical axis are missing.

Action taken: These have been updated.

R1.19. Fig. S16. What are the units of horizontal (Time) and vertical (MSD) axes? Please clarify.

Action taken: The caption has been updated: *“Note that distance and time are dimensionless units; the slope is invariant to the choice of units.”*

R1.20. In the Movies S1 and S2, I found several dust-like contaminations. I wonder what these contaminations are and how these contaminations affect the LM motion. I would appreciate the comments from the authors.

Movie S1 was not used for data analysis but was included because it clearly shows both a motile and immotile LM.

In Movie S2, we note the following:

None of the enframed areas had single motile LMs that were used in the analysis. We believe the marks to the left to be inherent scratches on the chips, while the rightmost box may enclose an aggregate of LMs (which, if so, is entirely immobile).

Reviewer #2

Korosec et al. describe a completely protein-based synthetic motor that operates under a burnt-bridge Brownian ratchet (BBR) mechanism. Specifically, they show that trypsin functionalized microspheres have directional diffusion on a peptide substrate lawn or 1D track. These results are a significant advancement in the field of synthetic motors because the authors use protein building blocks instead of DNA, which inherently have more diverse functions than DNA. Though the motor does not move in continuous motion, but rather moves in bursts and is frequently 'stalled', this is a first iteration and conceptually is very important to the field.

We are happy that the reviewer recognizes both the conceptual importance of the work and that a motor is not expected to be perfect in its first published iteration.

However, there is much improvement that needs to be made to the manuscript to better communicate the findings to a broad audience, such as the readership of Nature Communications. The text is bare bones, oftentimes leaving the reader to have to interpret what experiment/analysis is being performed directly from the Figures rather than being explained in the text. In addition, some of the claims need to be toned down. Lastly, very little detail is included characterizing the motor and track synthesis.

We thank the reviewer for their careful review and insightful comments. We have significantly expanded the description of methods and analysis throughout the text, and have addressed concerns with the claims. Specifics are provided in the responses that follow (and in those to Reviewer 1). The reviewer makes many excellent suggestions for follow-up studies. We fully agree with each of these and that our results open up many paths for future investigation,

highlighting the foundational character of our work. However, as we point out below, we feel that most of these suggestions are beyond the scope of this initial work but worthwhile pursuing in subsequent studies.

Specific Comments:

R2.1. Page 2, line 8: “Motion of a synthetic motor along a pre-defined track, a prerequisite for many potential nanotechnological applications, has to our knowledge not yet been realized.” Many of the motors cited throughout the paper operate along a pre-defined track.

By “track”, we mean to imply a one-dimensional path rather than a two-dimensional surface. We were also trying to draw analogies to the microns-long cytoskeletal and extracellular filaments along which biological motors operate. To our knowledge, motion along such tracks has not been demonstrated by synthetic motors. However, we appreciate the confusion that could arise from this wording and so have modified the sentence accordingly.

Action taken: We have rephrased the text to read

“While impressive, these designs have not yet achieved motion along lengthy one-dimensional tracks, akin to cytoskeletal and extracellular filaments and a prerequisite for many potential nanotechnological applications.”

R2.2. Page 2, line 14: “but to our knowledge, motility of a synthetic motor constructed of proteins – the material system that enables the complexity of life – has yet to be demonstrated”. The authors cite a paper in the supplemental describing such a motor. Should reword to say is an ‘important ongoing challenge’ to be more accurate. In addition, the authors should add this discussion in the main text, comparing and contrasting the motors, and properties.

We cannot determine which motor the reviewer is referring to, that we cited in the supplemental information. If this is our previously reported design and synthesis of the quantum-dot-based lawnmower, we did not investigate its motility in that work nor since. This present manuscript is the first to present an investigation of the motor-like dynamics of the lawnmower and, to our knowledge, the first to present experimental results on the motion of any synthetic, protein-based motor.

Action taken: We have nonetheless changed the wording as suggested on p. 2. Following the reviewer’s request, we now write

“but to our knowledge, motility of a synthetic motor constructed of proteins – the material system that enables the complexity of life – is an important ongoing challenge.”

R2.3. Page 2, line 29: “Many different biological systems undergo directed motion as BBRs (28-33), the mechanism of choice for regulated enzymatic degradation of key extracellular scaffolds in animals and plants (34-38).” List some of these biological systems, rather than making the reader look up these sources.

Action taken: We have provided a list of example BBRs in biological systems, with accompanying references:

“Many different biological systems undergo directed motion as BBRs. These include the Influenza virus (28-30), bacterial plasmid partitioning (31) and bacterial engulfment (32), and the ligand-depleting migration mechanism of metazoan cells (33). The BBR mechanism appears to be the mechanism of choice for regulated enzymatic degradation of key extracellular scaffolds in animals and plants, including collagen-degrading mammalian matrix-metalloproteases (MMPs) (34, 35), cellulase (36, 37) and chitinase (38, 39).”

R2.4.a. The details regarding motor synthesis and track synthesis and characterization are severely lacking.

Action taken:

We have added a detailed, self-contained Methods section to the main text, providing step-by-step protocols for each aspect of the work presented in this manuscript (e.g. motor synthesis, track/lawn preparation, surface functionalization, characterization).

R2.4.b. For the motor synthesis, the authors simply cite Figure S1, which has the wrong chemical structure for the maleimide.

Action taken:

We have corrected the chemical structure in Figure S1B – thank you to the reviewer for this careful reading of the manuscript and catching this error! We have additionally added a detailed section to the Methods on LM synthesis.

R2.4.c. For example, how many motors are attached. Is the system even multivalent.

Action taken:

We have performed further work to estimate the polyvalency of the LM. Briefly, our experiments estimate $(5 \pm 1) \times 10^5$ active trypsins per lawnmower, or about 0.02 per nm^2 of the surface area of the microsphere. The density of the PEG chains - and hence peptides - on the surface is estimated as 0.1 / nm^2 . Thus, the peptides are at a 5X greater density than the trypsins. The main text has been updated as follows:

“We verified the activity of LMs in solution and compared their peptide cleavage rate to that of trypsin (Fig. S1). We estimate that LMs each accommodate $(5 \pm 1) \cdot 10^5$ active trypsins, or about 0.02 trypsins per nm^2 of the surface area of the microsphere. Of these, ~2000 trypsins can be engaged with the underlying peptide lawn, which presents $\sim 10^4$ peptides within this LM footprint (Supporting Text). Thus, LMs are active and highly polyvalent, with the potential for thousands of interactions with the lawn.”

We have also described this experimental determination in the methods section, the results are shown in Fig. S1 E, F, and the Supporting Text includes a deeper discussion of LM and lawn polyvalency.

R2.4.d. Also, there are very little to no details regarding the surface functionalization. Based on the methods, the authors are using a fluorescent substrate, but this is never described in the text, nor is this used to characterize the surface. Rather, there is a single figure (Fig S14) in the supplemental showing a fluorescent image of 'cleaved tracks', yet this is never cited in the main text.

Action taken:

We have added a section to the Methods containing sufficient information to reproduce the surface functionalization described in the cited surface chemistry manuscript (ref. 41).

We have used fluorescence imaging to demonstrate (in Figs. S13 and S14 and previously in ref. 41) that the peptides on the surface are accessible to trypsin cleavage: fluorescence increases following quencher release resulting from peptide cleavage by trypsin. In this present work, we track LM motion with brightfield microscopy, and thus the fluorescence is not used as a readout. This is now clearly stated in the Methods:

"In the current experiments, LM dynamics were tracked using bright-field microscopy, and fluorescence imaging was used only to confirm formation and accessibility of the peptide lawn ((41), Figs. S13, S14)."

R2.4.e. Also, figures S13-S16 are not cited in the text.

Action taken:

All supporting figures – including Figures S13-S15 – are now referenced in the main text. Specifically, Figs. S13-S15 are referenced from the Methods section. Figure S16 has been removed (see R1.7).

R2.5. How does immobilization affect trypsin activity? Can you please compare k_{cat} on surface using soluble substrate compared to k_{cat} free in solution? Also, please add a little more discussion about trypsin. There is a great body of literature describing its processive activity.

In our report on the best route to LM synthesis (ref 24), Kovacic screened many proteases for the retention of their activity following the chemical modifications required for tethering to amine-modified beads. Here we reproduce Figure 2 and Table 1 from that manuscript, and – as requested by the editor – attach a copy of the full manuscript to this submission.

Fig. 2. Activity of proteases after each chemical treatment step toward construction of lawnmower blades investigated using a fluorescence assay. Trypsin showed strong retention of activity after different chemical modifications.

TABLE I
INITIAL RATE OF CLEAVAGE FOR EACH PROTEASE FOLLOWING MODIFICATIONS, RELATIVE TO THE RATE OF THE UNMODIFIED CONTROL

Protease	TCEP	TCEP + biotin	Traut	Traut + biotin
Trypsin	0.76	1.25	0.73	0.74
Elastase	0.43	0.63	0.96	0.77
Pepsin	0.72	0.94	0.07	0.06
Thrombin	0.51	0.45	0.39	0.11

We found that trypsin was the best choice of protease due to its retention of activity following cysteine reduction via TCEP or via thiolation of its primary amines to thiols with Traut's reaction, followed by coupling to maleimide-PEG2-biotin.

Action taken: In the current manuscript we have added text to the Methods (Lawnmower synthesis) to describe the choice of trypsin and clarify the point about its activity:

“While the operational principle of the Lawnmower is agnostic to the type of protease used in its design, trypsin is used here because of its retention of catalytic activity following a variety of chemical treatments including reduction, thiolation of amines, and subsequent covalent coupling to its thiols (24).”

We have performed follow-up experiments comparing LM activity to that of trypsin free in solution. While we do not have a direct way of determining the effect of immobilization on trypsin activity (because we can't independently quantify trypsin density and activity on LMs), our results indicate strong trypsin activity of LMs (see also responses R1.2, R2.4).

Regarding the processivity of trypsin, we have been unable to find any literature that describes its activity as processive. By contrast, in the reports that we have found, trypsin is described as non-processive, contrasting its behaviour with the processive proteasome. One clear example of this contrast - and what is used to characterize the nonprocessive behaviour of trypsin - is described by Akopian, Kisselev and Goldberg in *J Biol Chem* 1997: **272**, 1791-1798 (DOI: <https://doi.org/10.1074/jbc.272.3.1791>).

Action taken: We have added text to address the processivity of trypsin within the Supporting Text's discussion of polyvalency, citing this reference:

"Because trypsin is a nonprocessive enzyme (7), the activity of a LM scales linearly with the number of tethered, active trypsins."

R2.6. Page 2, line 44 in supplemental: The authors mention in the SI that the surface functionalization 'The F127 brush of the bare lawn did an outstanding job of blocking nonspecific adhesion of the LMs to the surface (<1% immotile), consistent with its previously demonstrated blocking abilities with microspheres of other surface chemistries', yet the results show that these particles move much less on these surface compared to peptide surfaces. Can the authors please elaborate more on this difference, and add to main text when describing the motor and surface.

That the LMs move further on peptide lawns than on bare lawns is strong evidence of their motor-like motility. The LMs diffuse freely on the F127 bare lawns. If they could bind, but not cleave, the lawn, we would expect to see slower diffusion, decreasing as polyvalency increases (see ref 52). However, with the ability to cleave the peptides, and thereby move superdiffusively, they are able to traverse a significantly larger distance than that available from pure diffusion.

Actions taken:

We have added text to elaborate on this point:

"Although diffusion may be an effective means of exploring local space at short timescales, the coupling of chemical energy to directed motion allows molecular motors to travel much further at longer timescales than is possible under thermal diffusion. We see this motor-associated greater range of motion for peptide-fueled versus diffusing LMs clearly demonstrated even at our shortest observation times of 10 seconds, where the average distance travelled is $\langle \overline{\Delta r} \rangle = 580 \pm 200$ nm on peptide lawn versus the diffusive $\langle \overline{\Delta r} \rangle = 230 \pm 40$ nm on bare lawns."

We have also added a panel to Fig. S3 that shows an expanded view of the LM trajectories on bare lawn, demonstrating the diffusive nature of their motion. These are the same data as in Fig. 1B, but in Fig. 1B we plotted the trajectories on peptide and bare lawns over the same timescale (6.25 hours) and with the same spatial scaling of the plots, so that the distinction in range of travel was very clear. We have also added a panel to Fig. S2 that shows the LM trajectories on peptide lawn from Fig. 1B over their full measured timescale of 12.5 hours.

R2.7. Please be more quantitative rather than descriptive when discussing the results. For example, 'Over similar timescales LMs on peptide lawns travel much farther than on bare lawns.' How much further? Instead, describe results as, the motors reached dR of 6 μm , where for bare lawns, had dR of 0.6 μm .

The sentence to which the reviewer is referring comes very early in our presentation of results. We had intended this statement to be a qualitative statement based on what is visually apparent from the trajectories, to orient the reader to key distinctions that are readily visible in the initial trajectories before delving into the quantitative analysis.

Actions taken:

Text modified to convey the intent of this sentence:

"Qualitatively, it is immediately apparent that over similar timescales LMs on peptide lawns travel much farther than on bare lawns."

We have added a quantitative value for this later in the analysis:

"We see this motor-associated greater range of motion for peptide-fueled versus diffusing LMs clearly demonstrated even at our shortest observation times of 10 seconds, where the average distance travelled is $\langle \Delta r \rangle = 580 \pm 200 \text{ nm}$ on peptide lawn versus the diffusive $\langle \Delta r \rangle = 230 \pm 40 \text{ nm}$ on bare lawns."

R2.8. Page 3, line 10: 'A threshold value'- What is the value? How was this value determined? Please explain in text.

The MSD_TA values for each trajectory are shown in Fig. S4. The threshold is indicated on the plot and the value stated in the caption:

"Trajectories are classified as motile or non-motile based on a threshold of $\text{MSD}_{\text{TA}} = 10 \mu\text{m}^2$ at $\tau = 4400 \text{ s}$ (the longest time lag plotted and used for all MSD analysis of LMs on peptide lawns)."

Action taken: We have added this value to the main text:

"We classified a LM as "motile" if its mean-squared displacement exceeded a threshold value of $10 \mu\text{m}^2$ at $\tau = 4400 \text{ s}$, and "immotile" otherwise (Fig. S4A)."

R2.9. What is time interval for Figure 1C? It is mentioned in supplemental, but not main text.

Action taken: The caption for Figure 1C has been updated to provide this information:

"Step size Δr vs time throughout one LM trajectory on a peptide lawn. Δr is determined for each 10-second time interval."

R2.10. Fig. S5, why is one motor superdiffusive?

One motor has a much greater velocity than the others shown in Fig S5. However, Fig S4C shows that many motors are superdiffusive over the entirety of the 2D peptide experiment.

R2.11. Why does MSD-TA versus MSD-EA vary significantly. Can you explain why you switched analysis in text.

We have used both MSD-TA and MSD-EA to analyse the LM trajectories, and present the results of both analyses in the text. The ensemble-averaged MSD reports the mean squared deviation among the whole LM ensemble in absolute time, *i.e.*, relative to its starting time. The trajectory-averaged MSD instead is a measure of the MSD for each given trajectory as a function of time lag, *i.e.*, the mean of the squared deviations is determined for each time lag, across all such possible instances of a given time lag within the data. These two measures should agree in ergodic systems, in which taking an ensemble average should provide the same result as averaging over one trajectory, recorded for a very long time. As the reviewer notes, these two measures do not agree for LMs, and that is what we would expect. The LM is a non-ergodic system: its future behaviour depends on its history. If it cleaved a lawn in a certain region, it is less likely to revisit this region. Thus, the disagreement between MSD-TA and MSD-EA is evidence of the expected motor-like dynamics of the LM.

Action taken: We have added text to the paragraph that provides both MSD-EA and -TA analyses to describe why these do not agree:

“Importantly, the difference between MSD_{EA} and MSD_{TA} indicates that the system is nonergodic (45). This is consistent with the history-dependent dynamics of LMs: their trajectories are influenced by what regions of the peptide lawn they have previously visited and cleaved.”

R2.12. Figure 1E: What exactly is $P(\Delta r)$. Please define/describe in text. This is an example where the reader has to interpret what analysis is being performed. Also, please just show Inset for figure 1E. Can move the ‘combined’ as a supplemental figure.

$P(\Delta r)$ represents the probability distribution of displacements Δr observed within a trajectory.

Actions taken:

$P(\Delta r)$ is defined when it is first mentioned:

“The enhanced motility of LMs on peptide lawns is also revealed by the probability distribution of their displacements $P(\Delta r)$ and corresponding speeds v , measured within each $\Delta t=10$ s recorded time interval.”

Following the reviewer’s suggestion, we have changed the presentation of results in Figure 1. We feel that both the main panel and previous inset of Figure 1E are important to include in the main text, and so we have put each into its own panel within Figure 1, so that they are larger and easier to view. Fig 1E shows the contrast of *all* LMs on peptide lawn with bare lawn, while Fig 1F plots separately the displacement distributions for the motile and immotile classes of LMs on peptide lawn.

R2.13. Figure 1F: Why is there a larger distribution showing high speeds, when motion is described mostly as saltatory? Also, the authors mention that the motor is oftentimes stalled,

which underestimates alpha, so why is the largest bin the fastest speeds. This doesn't agree with Fig. 1c.

Fig. 1F (now Fig. 1G) presents the distribution of individual LM speeds, where each LM's speed \bar{v} is obtained from the average of its displacements Δr divided by the time interval $\Delta t = 10$ seconds for which the Δr was determined. Within a given trajectory, the dynamics are saltatory, and thus, this average speed of a given LM \bar{v} is highly dependent on the amount of time it spends in a motile vs stalled state. The same is true for the complementary measure of individual LM performance α_{TA} : it is determined over the entire trajectory for each LM and thus combines contributions from motile and immotile dynamics.

Actions taken:

Equation 3 added to methods to show source of \bar{v} .

Text added to manuscript to describe this:

“Alternatively, for each LM we determined its average interval speed throughout its trajectory: $\bar{v} = \overline{\Delta r} / \Delta t$, where the average displacement per $\Delta t = 10$ sec interval $\overline{\Delta r}$ is given by Eq. 3. Even though LMs exhibit extended immotile dwells during their trajectories (Fig. 1C) that decrease their average speed \bar{v} , nonetheless there is a clear distinction between LM speeds when fuelled by peptides versus on a bare lawn (Fig. 2D).”

Text added to the caption of Fig. 1C that provides the average speed of this LM $\bar{v} = 36$ nm/s. We hope that this helps to clarify the source of this value, which contributes to the distribution in Fig. 1G (formerly 1F).

R2.14. Please describe Figure 2b in more detail. There looks like a very interesting result in bottom picture, fourth path from the right. Looks like there are two particles in same track, where the one particle runs into 'cleaved' substrate (i.e., product) and stops. What if you washed the surface and added particles again, would see a larger number of particles that stop because encountering 'cleaved' substrate? Also, what if you start the particles in a direction (due to gravity), then observed motion. Do all the particles move in the same direction. To do this experiment, the authors could make a 'binding buffer' with edta, to inactivate trypsin, then wash the surface with 'rxn buffer'? Lastly, Figure 2B shows same trajectories in S15, but Figure S15 has strange spots throughout? What are these spots? Can also see spots in Figure 2B, but less pronounced.

Figure 2B indeed shows the same trajectories as in Fig. S15. However, in Figure 2B, we show only the first image from the recording, while in Fig. S15, we overlay all images from the recording. Thus, Fig. S15 shows the LM trajectories in these experiments, including those that escape from the channels (not analysed). We believe these overlaid bead images are what the reviewer is referring to as "spots" in Fig. S15. The Methods section (Imaging and Tracking Lawnmower Motion) makes explicit which trajectories were used for analysis: *“Analysis of*

channel data considered only portions of trajectories that remained within a given channel (Fig. S15).” These are colour-coded in Fig. S15 and noted explicitly in the caption.

Action taken: We have updated the caption to Fig. S15 to clarify what is shown in the overlaid images:

“Overlay of all images in a LM channel experiment, where dark outlines show paths of LMs (sometimes escaping from a channel and diffusing into another). Trajectories included in Fig. 2 of the manuscript and our analysis are color-coded by their duration. These remain confined within a single track throughout the trajectory, and do not collide with other particles or ends of the channels.”

The reviewer proposes many excellent suggestions for further LM-channel experiments. We feel that they are beyond the scope of this current manuscript, which seeks to introduce our novel system on a two-dimensional substrate, then demonstrate its performance in a track-guided assay. In the concluding discussions of the manuscript, we mention many possible future directions for the work, aimed at further elucidating the underlying mechanism and optimizing performance; the reviewer’s suggestions are excellent as well.

Regarding the particle running into cleaved substrate, we have examined this trajectory and found that the LM does not stop but continues moving. This is clear in the following data plot:

R2.15. Should show Fig. 2c and 2d in same length scale.

Action taken: Figures have been updated to have the same x and y range for ease of comparison.

R2.16. What exactly is Fig. 2E? Should explain more in main text. Again, experiment and analysis is not clearly explained in text to non-expert.

We agree that Figure 2E was not presented in a clear stand-alone fashion. We have revised the figure and accompanying text to address this point.

Actions taken:

We have added a panel F to Figure 2 that shows the displacement distribution $P(\Delta x)$ for LMs in peptide channels (over a $\Delta t = 1$ second time lag – our measurement time interval for these data). This shows that the distribution is heavy-tailed: larger steps are more probable than predicted by a Gaussian distribution. The ratio of these distributions is plotted in Fig. 2E. Panel 2E also shows the analogous ratios between measured and Gaussian displacement distributions for particles in bare channels (Gaussian) and for the results of the simple 1D model (heavy-tailed). We see from these ratios that the peptide channels and the model both exhibit increasing weight at large displacements, while $P(\Delta x)$ on bare lawn channels is consistent with a Gaussian distribution (ratio $\cong 1$).

We have added a section in the Methods (“Comparison of experimental and model displacement distributions with Gaussian distributions”) to describe these calculations.

In the main text, we also added the specific values of kurtosis that were found for each $P(\Delta x)$ distribution, highlighting their distinct shapes:

“these LM displacement distributions $P(\Delta x)$ exhibit heavy tails. This is seen by the heightened probability of large displacements for LMs on peptide lawns relative to a Gaussian distribution, whereas displacements in bare lawn channels exhibit the expected Gaussian distribution associated with 1D diffusion (Fig. 2E, F).”

See also response R2.20.

R2.17. Page 5, line 26: ‘Microscale LMs have the potential for thousands of trypsins to be simultaneously engaged with peptides on the lawn.’ What if you vary trypsin density? How does this affect velocity/diffusion? Along the same lines, multivalency has been shown to improve processivity. Since your motors seem to have non-specific interactions with the surface (based on observing little motion on bare surfaces), it seems that multivalency isn’t needed? This would be a good section to compare to previous QD trypsin motor. I think this discussion and analysis is very important based on the body of literature of synthetic motors.

Varying trypsin density is another excellent suggestion for a future extension to this work. We would be very interested to investigate this in the future. Our simulations of multivalent random walkers (ref. 52) suggest that decreasing the multivalency will increase the diffusion coefficient. (See also response R1.3, above.)

We do not believe our motors have significant non-specific interactions with the F127 surface. By contrast, the diffusion coefficients recovered for their motion on these bare lawns agree very well with predictions (see Supporting Text). Thus, in the absence of peptide binding, the LMs are able to freely diffuse over the 2D surface.

Action taken: Text added to clarify this point:

“the observation of 2D diffusive motion is consistent with the previously demonstrated ability of the F127 surface to block nonspecific adhesion of these microspheres (41).”

Our previous work on the QD LM was a synthetic study, in which we demonstrated that trypsins on the QD retained their catalytic activity and thus that the LMs were capable of cleaving peptides. In it, we did not perform any experiments to measure their motility. A future comparison of LMs made with microscale hubs (as in the current study) and with QD hubs would be of interest and directly address the question of polyvalency. Similar questions have been addressed for DNA motors in a series of papers about the highly polyvalent DNA motor (HPDM) of the Salaita lab, where in different publications they varied the hub size from microscale to nanoscale (e.g. cited references 9, 10, 13, 15).

R2.18. A very important feature for BBM is that the motor has stronger affinity towards product compared to substrate. What exactly is this difference in affinity? Could you synthesize an ‘all product’ surface, and compare Kd’s. With DNA-based motors, this can easily be measured and compared.

It is important to clarify to the reviewer that a key to the BBR mechanism is a stronger affinity to substrate (as is the case for trypsin; see the added Ref. 51): a stronger substrate affinity drives a preferential binding to and subsequent cleavage of the substrate. Once it has been converted to product, the BBR preferentially binds to a new substrate. The comparison with dynamics on an all-product surface is another excellent suggestion for a follow-up study, but is beyond the scope of the current work. There would be many experimental complexities involved in that study, including the lengthy incubation periods to generate a product lawn, followed by multiple washes of the sample chamber to exchange its contents. Each wash brings a risk of surface delamination, and so this would be a nontrivial – but interesting and relevant – follow-up study.

R2.19. Add discussion comparing and contrasting motors attached to particle versus free in solution. For example, once protein is denatured, the motor could get stuck if immobilized on a particle. Where having added in solution has a continuous supply of motor.

Action taken: We have added to the discussion on this issue:

“Distinct from the related reconstituted Par and synthetic HPDM microscale BBR systems (14, 15), the LM does not rely on the supply of reagents from solution (Par proteins + ATP and RNaseH, respectively) to sustain motion. This offers the advantage of being a modular system that could be implemented in a variety of settings, without the need to maintain a supply of reagents. Sustained LM dynamics requires activity of its constituent trypsins; inactivated proteins cannot be replaced as easily as the solution-based supply in these other systems. In contrast to these solution-reliant BBRs, the free energy of the LM system is supplied within its prepared peptide lawn.”

R2.20. Page 5, line 5. What exactly is kurtosis? Please explain in more detail for the reader. For example, the authors just mention in text, and do not explain what this is or relate it to Fig.

2E. 'these LM displacement distributions exhibit a kurtosis $\kappa > 3$, indicating larger displacements than expected from a normal distribution (Fig. 2E). In contrast, particles on bare lawn have $\kappa \approx 3$ with displacements described by the expected Gaussian distribution for one-dimensional diffusive motion.'

Kurtosis is a parameter describing the shape - and more specifically the tailedness - of a distribution. It belongs to the same group of statistical parameters as variance and skewness, each with its own strict mathematical definition.

Actions taken:

We have expanded this paragraph to more clearly describe the analysis performed (relating also to R2.16):

"these LM displacement distributions $P(\Delta x)$ exhibit heavy tails. This is seen by the heightened probability of large displacements for LMs on peptide lawns relative to a Gaussian distribution, whereas displacements in bare lawn channels exhibit the expected Gaussian distribution associated with 1D diffusion (Fig. 2E, F). The shape of the displacement distribution can be quantified by its kurtosis (eq. 8), which indicates the tailedness of a distribution: for a Gaussian distribution, $\kappa = 3.0$. For LMs on peptide lawn channels, we find a kurtosis $\kappa = 3.2$, indicating larger displacements than expected from a normal distribution. In contrast, particles on bare lawn have $\kappa = 3.0$, the expected value for one-dimensional diffusive motion.

... Data modelled in this way show a kurtosis $\kappa = 3.2$, consistent with the heavy tails observed in experiments (Fig. 2E)."

We have also added a description of the kurtosis to the methods section, including a new equation (8).

See also response R2.16 for further information.

R2.21. Please elaborate on S11 analysis. I don't really see a difference.

Action taken: Text added to the caption of S11 to elaborate on the results of the analysis.

"The difference between the peptide and bare lawn channels is the pronounced dip in the histograms around the center (i.e. around scaled position 0), which is visible in all panels for the peptide channels but not the bare channels. More precisely, the bare channel histograms are consistent, within their statistical errors, with the hypothesis of a uniform histogram in each panel, because the difference in height between individual histogram bars is comparable to the size of the error bars. In contrast, for the peptide channels the difference in height between the histogram bars in the middle region and at the edges is several error bar sizes (and, moreover, is more systematic than for the bare channels), making it extremely unlikely that the peptide channel data is statistically consistent with a uniform histogram."

R2.22. Page 5, line 34. 'A heavy tail to the distribution of escape times from such a locally bound state can give rise to an MSD that scales sub-linearly with time, even for completely directional motion (Supplementary Text).' Wouldn't a solution be to look at larger times to tease out superdiffusion, greater than the 'stuck time'. By showing MSD vs time at various intervals, you should be able to observe the phase in the curve where directional diffusion is observed. For example, this is what was done for Fig. 1d green. So, fit that data, and report what range was analyzed in the text.

If the waiting time distribution is heavy-tailed, as proposed in this discussion, this implies that the mean waiting time is divergent. Thus, in this scenario it is impossible to wait out the "stuck time" - while some LMs would resume motility, others remain stuck. For such a heavy-tailed distribution, the effects persist into the asymptotic limit.

Action taken: With rationale following response R1.7, we have eliminated this discussion from the manuscript.

R2.23. What affects do salt and temperature have on motor properties? A potential major advantage of protein motors compared to DNA motors is minimal dependence on salt and temperature, comparatively. The authors should investigate this.

This is another fantastic suggestion for future follow-up work! We feel it warrants a stand-alone study, well beyond the scope of this current work that introduces and characterizes the motor on two-dimensional lawns and in channel-guided tracks, for the following reasons. Given the roles of salt and temperature in hydrophobic interactions - which are the basis for our lawn's adhesion to its underlying substrate - addressing this question requires significant effort in first studying the effects of salt and temperature on the lawn itself. This is certainly something we are interested in pursuing, but is beyond the scope of the current room-temperature study. We are unaware of any investigations of the temperature dependence of any BBRs (natural or synthetic); from a fundamental standpoint, this would be extremely interesting and highly relevant to providing a better understanding of this unique motor mechanism.

REVIEWER COMMENTS

Reviewer #1 (Remarks to the Author):

The authors have addressed most of my previous concerns. Detailed descriptions of the Methods section are really helpful to understand the details of the experiments and data analyses. I also appreciate detailed characterizations of the LM and peptide lawn (multivalency, catalytic activity, and surface density). Having understood these details, while I highly evaluate the revised manuscript, I still have several major concerns especially about the motile and immotile states as described below.

1. In page 6 of the rebuttal letter, the authors describe that “We have attempted to perform within-trajectory segregation of states into motile and immotile, but the objective assignment of transient states proved very challenging in the context of the significant heterogeneity of dynamics within individual trajectories. We concluded that such separate analysis would be unreliable and raise questions about the details of the segregation.” Based on this conclusion, the authors analyzed entire trajectories of the LM on the 2D peptide lawn (Fig. 1). However, for the kurtosis analysis of the LM on the 1D peptide track (Fig. 2), the authors describe that “For these analyses, immotile periods for LMs in peptide lawn channels were disregarded and trajectories were truncated when LMs became immotile.” (page 11 line 18 of the manuscript). If the authors conclude that the separate analysis of motile and immotile states is unreliable, entire trajectories should be also used for the analysis of the LM on the 1D peptide track.

2. Related to comment #1, it is difficult for me to understand the intent of the description “Indeed, in some replicates LMs display increased periods of immotility in peptide channels and exhibit a kurtosis $\kappa \approx 3$ (Fig. S12)” (page 6 line 12). How does the $\kappa \approx 3$ relate to “subdiffusive, quasi-immotile behavior” (page 6 line 10) and “highly polyvalent systems” (page 6 line 11)? Furthermore, does this result mean that $\kappa \approx 3$ will be obtained if entire trajectories are analyzed? Please clarify these points.

3. If separate analysis is unreliable, how did the authors divide the single trajectory in Fig. 1C into different colored segments? If this is done subjectively, it should be clearly stated in the main text and the figure legend to avoid misunderstanding of the readers.

4. Related to comment #3, the authors describe that “We also see evidence of immotility coinciding with previously visited regions: for example, the long dwell from 4-6.5 hours indicated by the red colour in Fig. 1C overlays with the earlier portion of the trajectory indicated in blue (Fig. 1C, inset)” (page 6 line 15). Again, how did the authors identify the “immotility” periods?

5. If the immotile dwells last for hours, I do not think it is appropriate to describe them as “transient” (Fig. 1 C red line, page 3 line 25, page 6 line 20). I think the term “transient” is misleading to the readers, and recommend the authors not use this term when referring to the immotile dwells.

6. I do not agree with the authors’ statement “However, in contrast to these solution-reliant BBRs, the free energy of the LM system is supplied within its prepared peptide lawn and thus can be guided by this track” (page 7 line 12). The free energy is also supplied within its prepared RNA lawn in the case of the solution-reliant BBRs such as DNA/RNA/RNase H motors. Furthermore, these solution-reliant BBRs can be guided by the microfabricated RNA track. I think that how to supply enzymes (from surrounding

solution or bead surface) and how to guide the BBRs by using the micropatterned tracks are different things.

Reviewer #2 (Remarks to the Author):

Thank you to the authors for their response and clarifications. The text is improved with the added experimental details. Korosec et al. describe a protein-based motor that operates through a burnt bridge mechanism (BBM) and is comparable to similar sized motors that operate via BBM. However, this motor has different material components (i.e., enzyme-substrate pair) compared to other BB motors. The authors argue that this expands the potential for synthetic motors, which I agree. The authors describe two experiments, motor properties on a 2D lawn versus motor properties on a '1D' track. As expected, the motors have super diffusive properties. Surprisingly, the motors move in spurts of a motile state, which isn't that frequent. The authors explain that this behavior is due to the motor being enclosed by consumed substrate (i.e., product), but a detailed analysis is lacking. Another surprisingly property of LMs is that they have slightly more ballistic behavior compared to other motor designs. This is perhaps the most novel and interesting result. A potential explanation could be the motor architecture, as the motor is immobilized to the particle compared to HPDMs, but again, little discussion is made or investigations carried out. I believe carrying out this analysis (immotile state) and adding these discussions would greatly improve the manuscript. In addition, there are still some further clarifications, but these are relatively minor.

R2.1

I still fail to see the difference between previous synthetic motors. For example, please see (Nature Nanotechnology volume 9, pages39–43 (2014)) where Choi et al. studied a Dz-NP motor moving along a carbon nanotube track, which is similar to a cytoskeletal track. I suggest deleting this claim altogether or adding a clarifier:

While impressive, 'protein-based motors' have yet to achieve motion along...

R2.4a

How was the confidence interval measured in Figure S1? It is a bit surprising the two plots have similar confidence intervals when comparing how different the fits look. Figure S1F has many datapoints further away from the fit. Please include R2 in fit. Regardless of the measurement being noisy, the quantitative characterization of the motor particle is appreciated.

R2.4d

The authors have now added details regarding the peptide lawn in the method section. Thank you. Figure 1a should be more accurate and reflect that the peptide substrate is fluorescently labeled and has a quencher.

R2.5

The data the authors provide show that trypsin retains activity when chemically modified, but what about when immobilized to a large microparticle, that slows down diffusion, and limits the entropy. I understand this is a challenging characterization, so maybe add a disclaimer that the estimate is likely an overestimate. I strongly encourage the authors to think about this type of characterization in future work.

R2.5

For trypsin processivity discussion. Reference 7 is an article regarding DNAzymes. I am unsure how this relates to trypsin processivity ...Please see Nano Lett. 2012, 12, 7, 3793–3802. Upon thinking about this comment further, it may not be necessary to include this in discussion since enzyme is also immobilized and isn't comparing apples to apples.

R2.10

This response still doesn't answer the question. Yes, the motors are superdiffusive, but there is one motor with very distinctive, 'ballistic' activity. I suggest re-analyze this particle to see if there is a logical explanation, i.e., isotropic, exc. Or if this is simply an outlier.

R2.13

I am still confused how there are larger distributions at higher velocities. If the motors reside in a stalled state most of the time, indicated in the text (saltatory) and data in Figure S2, wouldn't the 0 nm/s be the largest bin? If only 'motile' velocities are being analyzed, please describe/define this in the caption. As in, what velocity threshold was determined to be 'immotile' and what velocities were analyzed.

R2.14

Please describe what dark 'spots' are in image in text. It wasn't initially clear that these are from the particle in a BF image. It looks like the motors have superdiffusive properties away from the tracks based on comparing trajectories FigS15A (lower image), B to Fig. S3A. Do the particles ever return back to the track, or once the particle exits the track, it rarely returns? What if you analyzed particle motion off the track and compare to motion on the track. I would expect a difference. I think these details and analysis would be interesting, rather than simply saying only motion on the track were analyzed.

R2.18

Sorry for the typo, yes higher affinity for substrate not product. Couldn't the authors simply buy cleaved substrate and functionalize the surface accordingly, and compare diffusive properties. Or convert the substrate to product with soluble trypsin. Due to substrate design, it should be straightforward to characterize converting substrate to product using fluorescence. Maybe this can explain why more

ballistic motion is observed compared to other BBMs. However, upon thinking about this more deeply, I do not think this is critical to the paper. However, it is an easy experiment to do.

Figure 1S: 'dark red dot' -> dark red square

We thank both reviewers for their careful reading of the revised manuscript. We respond fully to their comments below, and indicate below and in the marked-up copies of the manuscript and supporting information where we have made changes (indicated in red in those documents). We believe that these changes and responses address the concerns that were raised. Some comments were made that suggested potential future studies, and we reiterate below where we feel this is appropriate. As a field, we continue to learn about the operational principles of very well studied biological motors such as kinesin, and we understand that some questions cannot be answered by one study in isolation, but that the data should be presented to the community so that mechanistic models can be developed and deeper insight obtained in the future. We hope that the reviewers agree that ours is a manuscript that presents sound measurements and analysis of a *novel system*, with sufficient detail that others can understand and reproduce our results, and that this work will inspire many future studies in the field. We believe that the manuscript is highly appropriate for *Nature Communications*.

Reviewer #1:

The authors have addressed most of my previous concerns. Detailed descriptions of the Methods section are really helpful to understand the details of the experiments and data analyses. I also appreciate detailed characterizations of the LM and peptide lawn (multivalency, catalytic activity, and surface density). Having understood these details, while I highly evaluate the revised manuscript, I still have several major concerns especially about the motile and immotile states as described below.

We are glad that the reviewer found our revisions and further work to be helpful. As stated previously, we appreciated the careful comments of both reviewers and felt that they helped to improve the presentation of the work performed. The detailed mechanism responsible for the clearly visible motile and immotile states remains elusive; we have carried out additional analyses and revised the manuscript to present these along with a discussion of possible reasons for these dynamics, observed also in related systems (though thus far without a universal mechanism). We hope that the addition of a protein-based BBR to the literature will help the field to consider and develop testable models for these microscale universally observed saltatory dynamics.

R1.1. In page 6 of the rebuttal letter, the authors describe that “We have attempted to perform within-trajectory segregation of states into motile and immotile, but the objective assignment of transient states proved very challenging in the context of the significant heterogeneity of dynamics within individual trajectories. We concluded that such separate analysis would be unreliable and raise questions about the details of the segregation.” Based on this conclusion, the authors analyzed entire trajectories of the LM on the 2D peptide lawn (Fig. 1). However, for the kurtosis analysis of the LM on the 1D peptide track (Fig. 2), the authors describe that “For these analyses, immotile periods for LMs in peptide lawn channels were disregarded and trajectories were truncated when LMs became immotile.” (page 11 line 18 of the manuscript). If the authors conclude that the separate analysis of motile and immotile states is unreliable, entire trajectories should be also used for the analysis of the LM on the 1D peptide track.

We apologize for the confusion generated by our previous response. The reviewer is correct that the ability to segregate trajectories differs between the 2D and 1D lawns.

When the LMs enter an immotile state on the 1D lawn, they have a strong tendency to remain in that state (for the remainder of the experiment). As seen in Fig. 2C, these periods of immotility are qualitatively very obvious. Because of this, it was in the case of the 1D experiments straightforward to manually truncate the trajectories at a point where the displacements became negligible.

The situation is more complex for the 2D lawn, where immotility is often transient and the LM returns to motile dynamics following a dwell in an immotile state. As seen in the exemplary trajectory of Fig. 1C, the LM switches into and out of immotility throughout the 12 hours of measurement. Similar switching dynamics can be seen in other trajectories (e.g. Fig. S2). We have attempted to perform within-trajectory segmentation of motile and immotile states for the ensemble of two-dimensional LM trajectories on the peptide lawn. These attempts included thresholding by displacement and by standard deviation of displacements over different time windows. However, none of these objective criteria resulted in a satisfactory discrimination between motile and immotile states: there is a large amount of within-trajectory and ensemble heterogeneity that makes an objective criterion challenging to implement for all cases. Thus, rather than bias our analyses by an incomplete segregation, we feel it is more appropriate to analyse and present the data as we have done here.

Since the reliable, objective segregation of 2D trajectories proved intractable, we then followed the reviewer's suggestion and reanalyzed the 1D trajectories without truncation. We found that including the entire trajectory strongly influenced the resulting displacement distribution. Inclusion of the immotile state in this analysis unsurprisingly enhanced the probability of observing displacements narrowly distributed about the origin:

We found that the full trajectory displacement distribution was not well described by a Gaussian distribution:

Not surprisingly, the addition of a narrow central peak resulted in an inability to describe the full width with a Gaussian distribution, resulting in a kurtosis of the complete distribution of $\kappa=4.0$, substantially larger than for the truncated trajectory distribution ($\kappa=3.3$).

Actions taken:

The manuscript has been amended to include these results as follows.

Main text, p. 5:

“these LM displacement distributions $P(\Delta x)$ exhibit heavy tails, even when immotile dwells are excluded.”

“Because this analysis truncated trajectories when they became immotile, the heavy tails of the distribution indicate motor-driven motility (Methods and Supporting Text).”

Main text, Analytical methods, p. 11:

“For the analyses presented in the main text, immotile periods for LMs in peptide lawn channels were disregarded and trajectories were truncated when LMs became immotile. We also explored the effects of including and excluding these qualitatively determined periods of immotility from the kurtosis analysis, and the effect of sampling at different time intervals (Fig. S16 and Supporting Text).”

Figure S16 and caption have been added to present the distributions without (Fig. 2F results) and with including immotile dwells. The figure is as shown above. The caption reads as follows:

“Fig. S16. *Comparison of displacement distributions of LMs in channels for $\tau = 1$ second intervals, for trajectories truncated when they become immotile (top; same result as presented in Fig. 2F) and for entire trajectories (bottom). Immotile dwells contribute a narrow peak centered at $\Delta x = 0$. Consequently, the kurtosis of the distribution is increased and a Gaussian fit fails to capture the full width of the distribution. Distribution statistics are as follows. Top plot: mean displacement = $0.0086 \mu\text{m}$, $\sigma = 0.334 \mu\text{m}$, $\kappa = 3.28$. Bottom plot: mean displacement = $0.0066 \mu\text{m}$, $\sigma = 0.298 \mu\text{m}$, $\kappa = 4.00$. Gaussians are plotted using the mean and standard deviation of their respective distribution.”*

We also evaluated the effects of sampling interval on kurtosis, finding values of $\kappa > 3$ for the three sampling times investigated. We have added a description of all of these findings and their implications to the Supporting Text, in a section entitled “Kurtosis”:

“A value of kurtosis above 3.0 for the displacement distributions indicates increased probability of large displacements relative to the central portion of the distribution. For LMs in channels, we found this could result from two possible modifications: (i) increased probability weight around $\Delta x = 0$; and (ii) increased probability weight in the tails of the distribution. For LMs, effect (i) results from including immotile segments, which contribute a narrow peak centered at $\Delta x = 0$ (Fig. S16). Effect (ii) results from motor activity, as seen in our 1D model results. Thus, when removing the effect (i) by excluding immotile regions from trajectories, any findings of kurtosis > 3 should indicate motor-associated motility. This was what we found when analyzing trajectories that we truncated once they appeared immotile.

The motile displacement distribution differs significantly from a Gaussian only in its wings (Fig. 2E). By contrast, for the immotile-included distribution shown in Fig. S16, the ratio of experimental:Gaussian probabilities around zero is significantly greater than one (arising from the contribution of the immotile state).

We found the kurtosis of $\kappa > 3$ to be unaffected by sampling interval. We repeated the analysis of these same motile-only truncated trajectories that had been sampled with $\tau = 1$ s, using longer sampling intervals of $\tau = 5$ s and $\tau = 10$ s, obtaining for these $\kappa = 3.14$ and $\kappa = 3.17$, respectively, slightly smaller values than the kurtosis of $\kappa = 3.28$ found for $\tau = 1$ s.”

R1.2. Related to comment #1, it is difficult for me to understand the intent of the description “Indeed, in some replicates LMs display increased periods of immotility in peptide channels and

exhibit a kurtosis $\kappa \approx 3$ (Fig. S12)” (page 6 line 12). How does the $\kappa \approx 3$ relate to “subdiffusive, quasi-immotile behavior” (page 6 line 10) and “highly polyvalent systems” (page 6 line 11)? Furthermore, does this result mean that $\kappa \approx 3$ will be obtained if entire trajectories are analyzed? Please clarify these points.

We naively thought that immotile periods would narrow the distribution and hence reduce the kurtosis from > 3 to a value closer to, or even below, 3. However, the analysis performed above (response R1.1) convinced us that this naïve view was incorrect.

Action taken:

The manuscript has been modified to remove reference to kurtosis and immotility in the context the reviewer mentions here. The effect of immotility on kurtosis is now included, as described in response R1.1.

R1.3. If separate analysis is unreliable, how did the authors divide the single trajectory in Fig. 1C into different colored segments? If this is done subjectively, it should be clearly stated in the main text and the figure legend to avoid misunderstanding of the readers.

We apologize for omitting this detail. The referee is correct: we conducted this analysis on the trajectory of Fig 1C based on visual inspection.

Actions taken:

Text added to Fig. 1C caption:

“Different regions have been colored based on visual inspection, as a guide to the dynamical heterogeneity within a trajectory.”

Phrasing updated in the main text, p. 6:

“the long dwell from 4-6.5 hours indicated illustrated by the red colour in Fig. 1C overlays with the earlier portion of the trajectory indicated shown in blue (Fig. 1C, inset)”

R1.4. Related to comment #3, the authors describe that “We also see evidence of immotility coinciding with previously visited regions: for example, the long dwell from 4-6.5 hours indicated by the red colour in Fig. 1C overlays with the earlier portion of the trajectory indicated in blue (Fig. 1C, inset)” (page 6 line 15). Again, how did the authors identify the “immotility” periods?

This was done manually.

Action taken: See previous response R1.3.

R1.5. If the immotile dwells last for hours, I do not think it is appropriate to describe them as “transient” (Fig. 1 C red line, page 3 line 25, page 6 line 20). I think the term “transient” is misleading to the readers, and recommend the authors not use this term when referring to the immotile dwells.

We intended “transient” to convey that the dwells are impermanent, but recognize that this phrasing may convey a different meaning of response to a perturbation.

Actions taken:

We have changed the wording as follows:

p. 3, line 25 from “while they may contain transient immotile dwells” to

“while they may contain periods of immotility ranging from minutes to hours”

p. 6, line 20 from “A second mechanism for transient immotility in highly polyvalent systems” to

“A second mechanism for temporary, often long-duration, immotility in highly polyvalent systems”

R1.6. I do not agree with the authors’ statement “However, in contrast to these solution-reliant BBRs, the free energy of the LM system is supplied within its prepared peptide lawn and thus can be guided by this track” (page 7 line 12). The free energy is also supplied within its prepared RNA lawn in the case of the solution-reliant BBRs such as DNA/RNA/RNase H motors. Furthermore, these solution-reliant BBRs can be guided by the microfabricated RNA track. I think that how to supply enzymes (from surrounding solution or bead surface) and how to guide the BBRs by using the micropatterned tracks are different things.

This is an excellent point and we welcome the opportunity to clarify the issue.

Action taken:

Wording has been changed to

“However, in contrast to these solution-reliant BBRs, the LM system does not require external reagents to be provided to maintain activity.”

Reviewer #2:

Thank you to the authors for their response and clarifications. The text is improved with the added experimental details. Korosec et al. describe a protein-based motor that operates through a burnt bridge mechanism (BBM) and is comparable to similar sized motors that operate via BBM. However, this motor has different material components (i.e., enzyme-substrate pair) compared to other BB motors. The authors argue that this expands the potential for synthetic motors, which I agree. The authors describe two experiments, motor properties on a 2D lawn versus motor properties on a ‘1D’ track. As expected, the motors have super diffusive properties. Surprisingly, the motors move in spurts of a motile state, which isn’t that frequent. The authors explain that this behavior is due to the motor being enclosed by consumed substrate (i.e., product), but a detailed analysis is lacking. Another surprisingly property of LMs is that they have slightly more ballistic behavior compared to other motor designs. This is perhaps the most novel and interesting result. A potential explanation could be the motor architecture, as the motor is immobilized to the particle compared to HPDMs, but again, little discussion is made or investigations carried out. I believe carrying out this analysis (immotile state) and adding these discussions would greatly improve the manuscript. In addition, there are still some further clarifications, but these are relatively minor.

We are glad to read that the reviewer feels that our revisions improved the text, and that the work expands the potential for synthetic motors. The mechanism for the immotile state, observed also for other microscale BBRs, is of great interest to us as well. We have performed additional analyses to elucidate more details of the immotile state, and have included these and additional discussion in the manuscript about potential mechanisms for this state. Future work (by us and/or others) we hope may provide a unifying mechanism into what underlies the frequently observed saltatory dynamics in these novel motor systems. We have added text to the concluding discussion in the manuscript to convey this point. There are certainly always more questions that are posed from doing experiments, than can be answered in one study. We hope that the current study leads to many more investigations.

In our responses that follow, we keep with the reviewer's numbering, which refer to our original response document.

R2.1. I still fail to see the difference between previous synthetic motors. For example, please see (Nature Nanotechnology volume 9, pages39–43 (2014)) where Choi et al. studied a Dz-NP motor moving along a carbon nanotube track, which is similar to a cytoskeletal track. I suggest deleting this claim altogether or adding a clarifier:

While impressive, 'protein-based motors' have yet to achieve motion along...

Thank you for clarifying the motor to which your original comment referred. There are important distinctions between that work and ours, in terms of the building blocks and fabrication approaches. However, we absolutely agree that theirs is a track-based synthetic walker and should be noted explicitly as such!

Action taken:

The text has been modified to read

"With a notable exception (8), these designs have not yet achieved motion along lengthy one-dimensional tracks,"

where reference (8) is the 2014 paper in *Nature Nanotechnology* by Cha *et al.*

R2.4a. How was the confidence interval measured in Figure S1? It is a bit surprising the two plots have similar confidence intervals when comparing how different the fits look. Figure S1F has many datapoints further away from the fit. Please include R2 in fit. Regardless of the measurement being noisy, the quantitative characterization of the motor particle is appreciated.

It is important to note that the two plots in question differ significantly in scale and in the number of datapoints. The confidence interval for each value resulting from the linear $y(x)$ fit was calculated as follows:

$$CI_i = y_{i,predicted} \pm t \times SEP_i.$$

Here, t is the critical value from the t -distribution; t depends on the confidence level (0.95 here) and the degrees of freedom, calculated as $n - 2$, where n is the number of observations and

two parameters (slope, intercept) are estimated by the fit. SEP is the standard error of prediction:

$$SEP_i = \sqrt{MSE \times \left(\frac{1}{n} + \frac{(x_i - \bar{x})^2}{\sum_{i=1}^n (x_i - \bar{x})^2} \right)}.$$

Here, \bar{x} is the mean of the measured range ($\bar{x} = \frac{1}{n} \sum_{i=1}^n x_i$) and MSE is the mean squared error, calculated as

$$MSE = \frac{1}{n-2} \sum_{i=1}^n (y_i \text{ observed} - y_i \text{ predicted})^2.$$

The width of the confidence band is thus proportional to the standard error of the predicted value. For calculation of the confidence intervals, we used Origin, and verified the result using the above equations.

Action taken:

The value of R^2 for each linear fit is now included in the caption, which also now states the slope from Fig. S1E:

Fig. S1E: *Fluorescence increases linearly with time with rate $935 \pm 100 \text{ min}^{-1}$ ($R^2=0.95$).*

Fig. S1F: *The cleavage rate scales linearly with trypsin concentration ($R^2=0.62$).*

R2.4d

The authors have now added details regarding the peptide lawn in the method section. Thank you. Figure 1a should be more accurate and reflect that the peptide substrate is fluorescently labeled and has a quencher.

Because the work in the main text does not use fluorescence, we prefer to keep the schematic of Fig. 1A simple and showing just the key conceptual components. The methods section in the main text document provides detailed information on the composition of the peptide used to make the lawn.

Actions taken:

We have added the fluorophore and the quencher to the peptide in the more detailed Figure S1, where the fluorogenic properties of the peptide are exploited. The caption of Fig. S1D has been updated to include

“A fluorophore, fluorescein isothiocyanate (FITC), and quencher, 4-([4-(dimethylamino)phenyl]azo)benzoic acid (DABCYL), are located at the N and C termini of the peptide, respectively”

R2.5. The data the authors provide show that trypsin retains activity when chemically modified, but what about when immobilized to a large microparticle, that slows down diffusion, and limits the entropy. I understand this is a challenging characterization, so maybe add a disclaimer that the estimate is likely an overestimate. I strongly encourage the authors to think about this type of characterization in future work.

We agree that it would be a challenging characterization, but worth pursuing in future work that tests the effect of polyvalency on motor properties. For the purposes of this manuscript, we have clarified the point.

Actions taken:

We have added the following sentence to the end of the “Lawnmower polyvalency” section of the Methods, where this is described:

“This value represents the number of free trypsin equivalents: from previous work we learned that the activity of individual trypsins is unaffected by the chemical labelling performed here (24), but do not know how tethering to a large microsphere affects their activity.”

R2.5. For trypsin processivity discussion. Reference 7 is an article regarding DNAzymes. I am unsure how this relates to trypsin processivity ...Please see Nano Lett. 2012, 12, 7, 3793–3802. Upon thinking about this comment further, it may not be necessary to include this in discussion since enzyme is also immobilized and isn't comparing apples to apples.

The relevant reference about trypsin processivity is reference 7 in the Supporting Information, which is included in the main text reference list as reference 69. We think it is worthwhile to include this point, in case a reader is interested in processivity of trypsin.

R2.10. This response still doesn't answer the question. Yes, the motors are superdiffusive, but there is one motor with very distinctive, 'ballistic' activity. I suggest re-analyze this particle to see if there is a logical explanation, i.e., isotropic, exc. Or if this is simply an outlier.

We agree that the LM trajectory in question is an outlier, however, there are no timescales over which it is truly ballistic. This particular trajectory can be fully visualized as the brown trajectory in Fig. S2A, with the corresponding displacement trajectory $\Delta r(t)$ in Fig S2B (brown, bottom right). Interestingly, this LM moved at an increased velocity (relative to all other LMs) for only approximately 20 minutes at the beginning of the experiment, before the speed took on values similar to the other peptide-driven LMs. The anomalous diffusion coefficient over the first 20 minutes is approximately 1.7, while for the entire trajectory it is 1.2 (and is thus not one of the more superdiffusive trajectories relative to the entire ensemble throughout the entire experiment). We believe it is an oddly behaving outlier. With the displacement trajectories of a number of LMs provided in Fig. S2, the reader can get a sense of the variety of behavior observed.

R2.13. I am still confused how there are larger distributions at higher velocities. If the motors reside in a stalled state most of the time, indicated in the text (saltatory) and data in Figure S2, wouldn't the 0 nm/s be the largest bin? If only 'motile' velocities are being analyzed, please describe/define this in the caption. As in, what velocity threshold was determined to be 'immotile' and what velocities were analyzed.

The figure panel from which this comment originated is Fig. 1G, which shows the average speed of each LM throughout its 12.5-hour measurement time. During this time, “motile” LMs are likely to exhibit periods of motility and immotility. If we consider the LM of Fig. 1C, its average speed is 36 nm/sec (see caption) and contributes to the histogram of green (motile) LM speeds in Figure 1G at this value. Looking at the displacement trajectory in Fig. 1C, it is clear that there are many time intervals within the 12.5-hour trajectory in which the LM exhibits a much greater speed, traversing $>1 \mu\text{m}$ in a 10-second interval ($>100 \text{ nm/s}$). Fig. 1G shows the distribution of average speeds, obtained for each LM using equation (3). Thus, the number of points contributing to the histogram is the number of LMs measured.

If we instead look at the *distribution of speeds within* the LM trajectories, then we see a distribution peaked at the $v = 0$ bin as expected (equivalently, $\Delta r = 0$ displacement over the 10-second measurement window): Fig. 1E, Fig. 1F. This is because of the immotile dwells. A maximum at $\Delta r = 0$ is not seen for LMs on bare lawns (Fig. 1E, 1F) because in 2D diffusion, it is highly unlikely that the particle does not move. Instead, the distribution is peaked away from zero, and is well described by the Rayleigh distribution (equation 1 and Fig. S8).

R2.14. Please describe what dark ‘spots’ are in image in text. It wasn’t initially clear that these are from the particle in a BF image. It looks like the motors have superdiffusive properties away from the tracks based on comparing trajectories FigS15A (lower image), B to Fig. S3A. Do the particles ever return back to the track, or once the particle exits the track, it rarely returns? What if you analyzed particle motion off the track and compare to motion on the track. I would expect a difference. I think these details and analysis would be interesting, rather than simply saying only motion on the track were analyzed.

We identified these dark spots in our previous response but neglected to update the caption to Fig. S15 at the time. Apologies.

Action taken:

Caption to Fig. S15 updated to note

“LMs appear as dark circles in the brightfield images, and dark outlines show the paths of LMs”

Outside channels, there is neither peptide-functionalized nor bare lawn as these areas are not hydrophobic after TMCS treatment and thus do not allow for attachment of the F127 brush. Therefore, this motion is on a different type of surface and is not directly comparable to that in channels. For this reason, we have not characterized the motion outside of the channels. The trajectories shown in Fig. S3A exemplify 2D diffusion, rather than superdiffusion, as they are on a bare F127 lawn and are well described by the Rayleigh distribution. Some LMs do return to the channels after jumping out, while others jump into a neighboring channel. We analyze those portions of the trajectories that occur within a given channel.

R2.18. Sorry for the typo, yes higher affinity for substrate not product. Couldn’t the authors simply buy cleaved substrate and functionalize the surface accordingly, and compare diffusive

properties. Or convert the substrate to product with soluble trypsin. Due to substrate design, it should be straightforward to characterize converting substrate to product using fluorescence. Maybe this can explain why more ballistic motion is observed compared to other BBMs. However, upon thinking about this more deeply, I do not think this is critical to the paper. However, it is an easy experiment to do.

We agree that this would be a clear extension to the project and provide insight as to the mechanism. We also agree that it is not critical to the paper, and don't have a set of trained hands in the lab that could undertake this measurement at the present time.

Figure 1S: 'dark red dot' -> dark red square

Thank you for the eagle eyes!

Action taken:

We have fixed the labelling in the caption of Fig. S1, E and F.

REVIEWERS' COMMENTS

Reviewer #1 (Remarks to the Author):

The authors addressed all of my concerns in the second revision.

Reviewer #2 (Remarks to the Author):

Essentially, the authors describe LM dynamics on 2D surfaces (Fig. 1) and 'pseudo-1D' tracks (Fig. 2) and show that LMs have biased motion and increased velocities compared to randomly diffusing particles (when in the motile state). Though this is not surprising, it is a different class of motor that potentially expands the capabilities of BBR motors.

Overall, the authors have addressed my concerns. The clarifications in the text describing how motile motors were distinguished from immotile motors is much appreciated. Below are minor comments and I'll leave it up to the editor whether they are addressed.

1. This statement is not accurate. 'track-guided motion of a BBR motor has not yet been achieved'.

Please see:

- Nature. 2010 May 13; 465(7295): 206–210.;
- Nat Nano 11:184–35 190;
- Nature Nanotechnology volume 9, pages39–43 (2014)

2. Define metrics:

Pg 4. 'peptide lawns revealed by other metrics' ,

Reviewer #1 (Remarks to the Author):

The authors addressed all of my concerns in the second revision.

We thank the reviewer for their time and careful consideration of our work throughout the review process.

Reviewer #2 (Remarks to the Author):

Essentially, the authors describe LM dynamics on 2D surfaces (Fig. 1) and 'pseudo-1D' tracks (Fig. 2) and show that LMs have biased motion and increased velocities compared to randomly diffusing particles (when in the motile state). Though this is not surprising, it is a different class of motor that potentially expands the capabilities of BBR motors.

Overall, the authors have addressed my concerns. The clarifications in the text describing how motile motors were distinguished from immotile motors is much appreciated. Below are minor comments and I'll leave it up to the editor whether they are addressed.

1. This statement is not accurate. 'track-guided motion of a BBR motor has not yet been achieved'.

Please see:

- Nature. 2010 May 13; 465(7295): 206–210.;
- Nat Nano 11:184–35 190;
- Nature Nanotechnology volume 9, pages39–43 (2014)

This statement indeed should be rephrased, which we have now done. It now reads "However, to our knowledge, long-range track-guided motor activity of a polyvalent BBR has not yet been achieved."

The qualifications are important, as follows.

In the 2010 *Nature* publication, Lund et al. showed that DNA-based molecular "spiders" could cleave substrates positioned on DNA origami at specific sites. Visualizing the motion in real time was extremely challenging, but spiders appeared to exhibit biased motion. The range of motion was <100 nm, and hence not long-range.

The *Nature Nanotechnology* paper of Cha et al. in 2014 is the sole example of long-range (micrometer scale) directed motion of a BBR. In our previous revision, we modified the first paragraph of the introduction to note this success explicitly ("With a notable exception (8), these designs have not achieved motion along lengthy one-dimensional tracks"). There are two important distinctions between the BBR of Cha et al. and our LM: their study was of a monovalent BBR and it was a DNAzyme. These features contrast with our polyvalent BBR, whose activity derives from protein-base enzymes.

The DNA-based HPDMs of the 2015 *Nature Nanotechnology* paper by Yehl et al. bear the greatest similarity to the Lawnmower, and we compare and contrast these systems extensively in our manuscript. Yehl et al. attempted to improve motor activity by providing patterned quasi-1D tracks for the HPDMs. However, they did not see an improvement in activity, which they ascribed to two main potential reasons: “Many particles became entrapped, partially because of RNA cross-contamination into the PEG-passivated regions (Fig. 4c) and partially because of self-entrapment in consumed substrate corrals.” We believe that our patterning of peptide tracks avoids the cross-contamination issue.

We have clarified these important distinctions later in the manuscript, where we now include a comparison in our discussion (p. 6):

“Indeed, nucleic-acid-based BBRs of low polyvalency followed patterned tracks (over short distances (7) and over micrometers (8)), while optimization of motor-like motility along tracks for highly polyvalent systems such as HPDMs (16) and the LM remains a future challenge.”

2. Define metrics:

Pg 4. ‘peptide lawns revealed by other metrics’

Because we refer in this statement “...peptide lawns revealed by other metrics (Figs. 1C, S2, S4)” specifically to the data, we have opted to keep a smooth flow of text by not listing each of these metrics separately in this phrase. Instead, we have captured the basis of these by stating “...peptide lawns revealed by other displacement-based metrics (Fig. 1C, Supplementary Figs. 2, 4)”.